# Programmable Energy-Efficient Analog Multilayer Perceptron Architecture Suitable for Future Expansion to Hardware Accelerators

**Jeff Dix** [1] , **Jeremy Holleman** [2] **and Benjamin J. Blalock** [3,*]

1   Electrical Engineering Department, University of Arkansas, Fayetteville, AR 72701, USA; dix@uark.edu
2   Electrical and Computer Engineering Department, University of North Carolina at Charlotte, Charlotte, NC 28262, USA; jhollem3@uncc.edu
3   Department of Electrical Engineering and Computer Science, University of Tennessee at Knoxville, Knoxville, TN 37996, USA
*   Correspondence: bblalock@utk.edu; Tel.: +1-479-575-6051

**Abstract:** A programmable, energy-efficient analog hardware implementation of a multilayer perceptron (MLP) is presented featuring a highly programmable system that offers the user the capability to create an MLP neural network hardware design within the available framework. In addition to programmability, this implementation provides energy-efficient operation via analog/mixed-signal design. The configurable system is made up of 12 neurons and is fabricated in a standard 130 nm CMOS process occupying approximately 1 mm$^2$ of on-chip area. The system architecture is analyzed in several different configurations with each achieving a power efficiency of greater than 1 tera-operations per watt. This work offers an energy-efficient and scalable alternative to digital configurable neural networks that can be built upon to create larger networks capable of standard machine learning applications, such as image and text classification. This research details a programmable hardware implementation of an MLP that achieves a peak power efficiency of 5.23 tera-operations per watt while consuming considerably less power than comparable digital and analog designs. This paper describes circuit elements that can readily be scaled up at the system level to create a larger neural network architecture capable of improved energy efficiency.

**Keywords:** neural network; multilayer perceptron; energy efficient; analog; weak inversion; programmable

## 1. Introduction

Artificial neural networks (ANNs) are machine learning architectures that are inspired by biological neural structures and can perform several tasks with an example being function approximation [1]. The MLP is an early type of neural network that classifies data with nonlinear functions in parallel signal pathways [2]. The MLP structure consists of an input layer, a hidden layer, and an output layer. The input layer takes the set of data to be analyzed and passes it on to the hidden layer. The hidden layer may contain multiple layers that receive the outputs from the previous layer, multiply them with a weight, and output this weighted sum via a nonlinear activation function. Each layer consists of one or more neurons that create the overall ANN functionality. Two common nonlinear activation functions are the sigmoid and the hyperbolic tangent functions with the general function form:

$$y_i(\mathbf{x}) = \phi(\mathbf{w}_i'\mathbf{x} + b_i) = \phi(z_i) \tag{1}$$

where $\mathbf{x}$ is the vector of input values, $\phi()$ is the nonlinear activation function, $\mathbf{w}_i'$ is the vector containing the weight values, and $b_i$ is the bias for a given neuron. Figure 1 shows a general MLP structure with two hidden layers. A distinct characteristic of an MLP ANN is

that every neuron will output to all of the neurons of the subsequent layer and is typically used for the classification of a set of data that are not linearly separable [2].

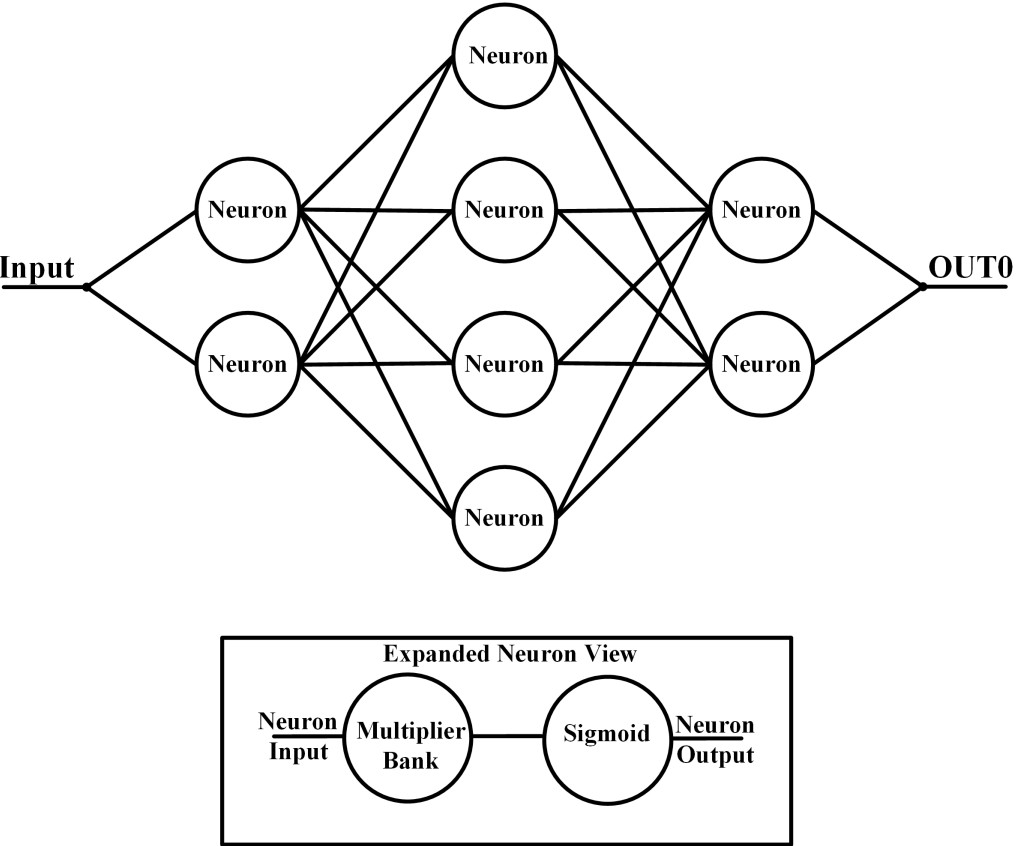

**Figure 1.** General MLP structure.

The complexity and power consumption of MLP hardware are critical performance characteristics that require optimization to produce the most efficient network possible for a specific problem. The implementation complexity can be addressed by utilizing either a field-programmable gate array (FPGA) or a digital signal processor (DSP) to easily and efficiently change the architecture of the MLP seamlessly. While these digital methods provide the programmability that designers desire, these systems do not efficiently optimize power consumption at the transistor level and can lead to increased system power consumption. Another method to add programmability to MLP hardware is to manually design circuit blocks on an integrated circuit (IC), providing better performance characteristics with greater transparency and control [3]. Overall, the programmability of an MLP hardware implementation is a key design parameter that allows the network to function in an optimal manner with the ability to turn off and on sections of the architecture without penalty to data throughput.

After considering the programmability of an MLP architecture, the overall power consumption of the network needs determination. The goal of any neural network circuitry is to provide the highest power efficiency, which is the number of computations per second per watt of power. Analog designs as a whole have proven over time that they are far more capable of low power computations (e.g., [4,5]). Analog offers design control over power dissipation at a transistor level while also taking advantage of transistor characteristics to optimize performance. Another power-saving aspect of analog circuits is that summation and subtraction operations are easily performed by a wire junction (current-mode circuits), whereas digital circuits require multibit subtractors and adders [6].

The ability to control the transistor's operation region is crucial in creating neural network hardware that has the basic functionality of an MLP with reduced power con-

sumption. Weak inversion provides these capabilities in two ways: exponential transfer characteristics and low power consumption [7]. Exponential transfer characteristics are vital to the successful operation of the hardware. The architecture's neurons, or nodes, require an activation function after the weight sum. In the case of MLP hardware, the activation function is a nonlinear operation and requires a special circuit design to produce the desired nonlinear functions [8]. With these characteristics in mind, an analog energy-efficient MLP design is highly desirable as it has the greatest potential to provide the highest number of computations per second for the lowest power consumption. The design of the programmable MLP hardware-based system is presented in Section 3, followed by the results, a brief discussion of the network results, and conclusions in Sections 2–5, respectively.

## 2. Results

This section details the measurement results obtained from physically testing the manufactured integrated circuit in the 130 nm technology node on the printed circuit board (PCB). The PCB is shown in Figure 2 and contains circuits to test the different elements of the system. The test board is divided into six sections: power circuits (red), microcontroller and supplemental circuitry (yellow), input signal circuits for four main inputs (purple), input signal circuits for simple neural network hardware system (orange), input circuits for test structures (black), and socket for integrated circuit (white). These six sections allow every circuit to be thoroughly tested and analyzed. All power rails for the integrated circuit are kept at 1.2 V, whereas the rails for the microcontroller are held at 3.3 V. The nominal power supply for the design is 1.2 V with a minimum testing frequency of 1 MHz (limited by the microcontroller-generated test inputs). The bit streams and input signals are generated by the microcontroller. A buffer circuit is used to obtain a cleaner output signal unless otherwise specified as the output structures of the chip are not strong enough to drive the 12 pF capacitive load of the oscilloscope probe at higher frequencies. In general, the PCB is capable of testing the system and other circuits at frequencies up to the ones of MHz. The frequency is limited to the low MHz range due to the chosen microcontroller and PCB parasitics within the signal pathway.

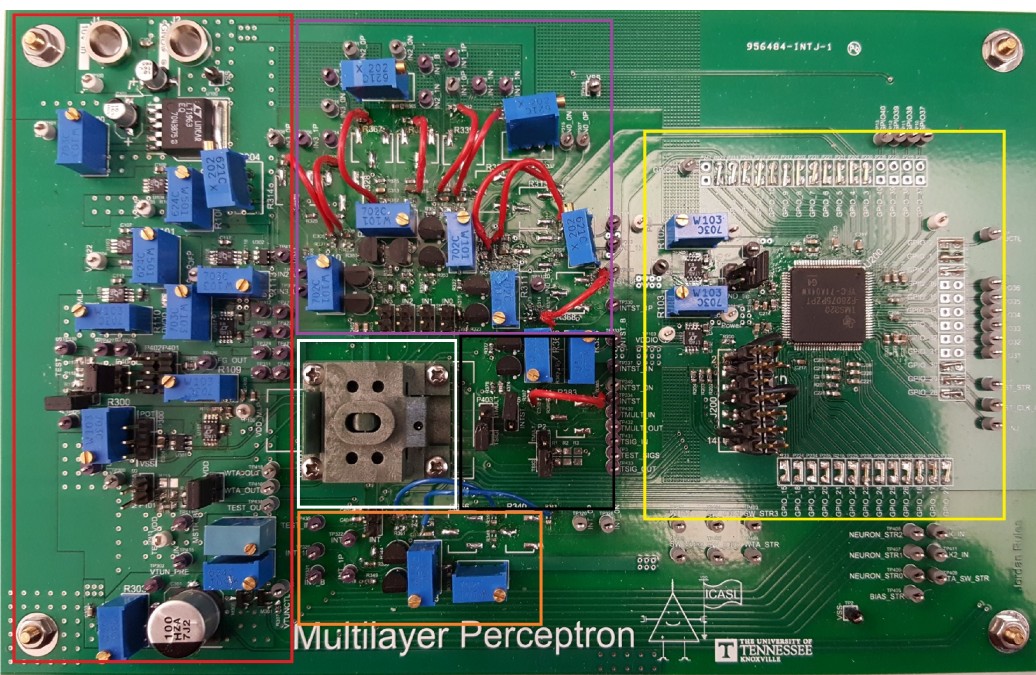

**Figure 2.** PCB test board for MLP chip (see text for color details).

The measurement results are for several different configurations for the system and a brief noise analysis on the first configuration. The measurement results utilize one or more inputs from the main input signal, generating a circuit section after the integrated circuit is programmed via the microcontroller. The programming consists of activating switches for the best routing scheme for each configuration and activating the bias cells to operate the hardware implementation at the highest possible frequency while still maintaining an accurate output signal. For all of the configurations, the architecture is programmed as a classifier to verify the programming and signal accuracy. The output results detail a basic classification operation where the system is providing a "high" output when the input signal(s) are "high" and "low" when the input signal(s) are "low". This simple operation offers the ability to effectively characterize each configuration's power and throughput for further analysis. Each configuration is judged on its ability to meet a high power efficiency of 1 tera-operations per watt (TOps/W), which is defined by Equation (2). Operations consist of either a sum or multiply operation within each activated neuron. In addition to the verification results for the configuration, the propagation delays for one of the rising and falling edges of the pulse train will be analyzed. All of the configuration will have input voltage signals that have a minimum of 200 mV and a maximum in the range of 900 to 1.1 V, depending upon the load on the microcontroller's IO ports (lower for higher number of inputs to the chip). Lastly, the input signal is operating at the worst-case situation, which is alternating between high and low states.

$$Power\ Efficiency = \frac{(operations)(frequency)}{Power} \tag{2}$$

Figure 3 depicts the first configuration analyzed for measurement results with one input and three neurons. Figure 4 details the input voltage waveform in yellow and the output voltage waveform in green for this first configuration. The frequency for the input signal data is 4.06 MHz, which is reflected in the output waveform that goes from a low state at zero to a high state at 1.2 V. The measured on-state current for this configuration is 14 µA with the off-state current being 7 µA. These currents averaged over the 8 on states and 8 off states of the signal and multiplied with the voltage give an average power of 12.6 µW. This configuration has 6 total operations (2 per neuron with 3 neurons). Therefore, the power efficiency is 1.93 TOPS/W for this configuration and input signal. Figures 5 and 6 show the propagation delay for the rising and falling edges, respectively. The rising edge propagation delay is 338 ns, while the falling edge is 489 ns. The large delays come from the thresholding circuitry as it requires large currents to change the voltage signal at the comparison node and the numerous parasitics encountered from the routing and switches.

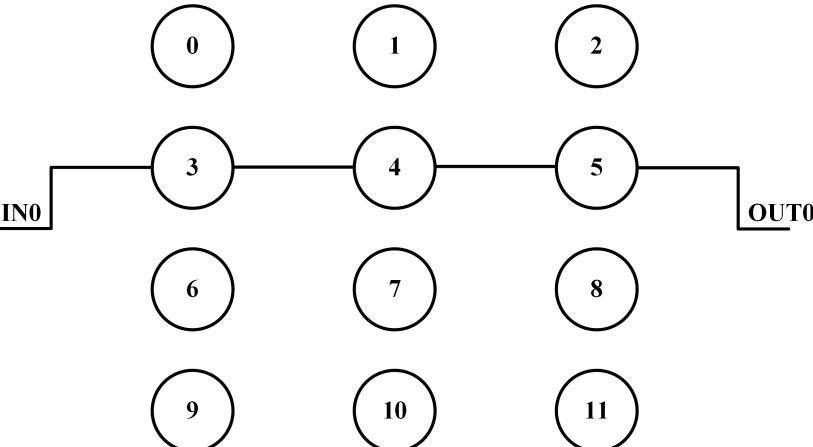

**Figure 3.** Hardware configuration for first configuration.

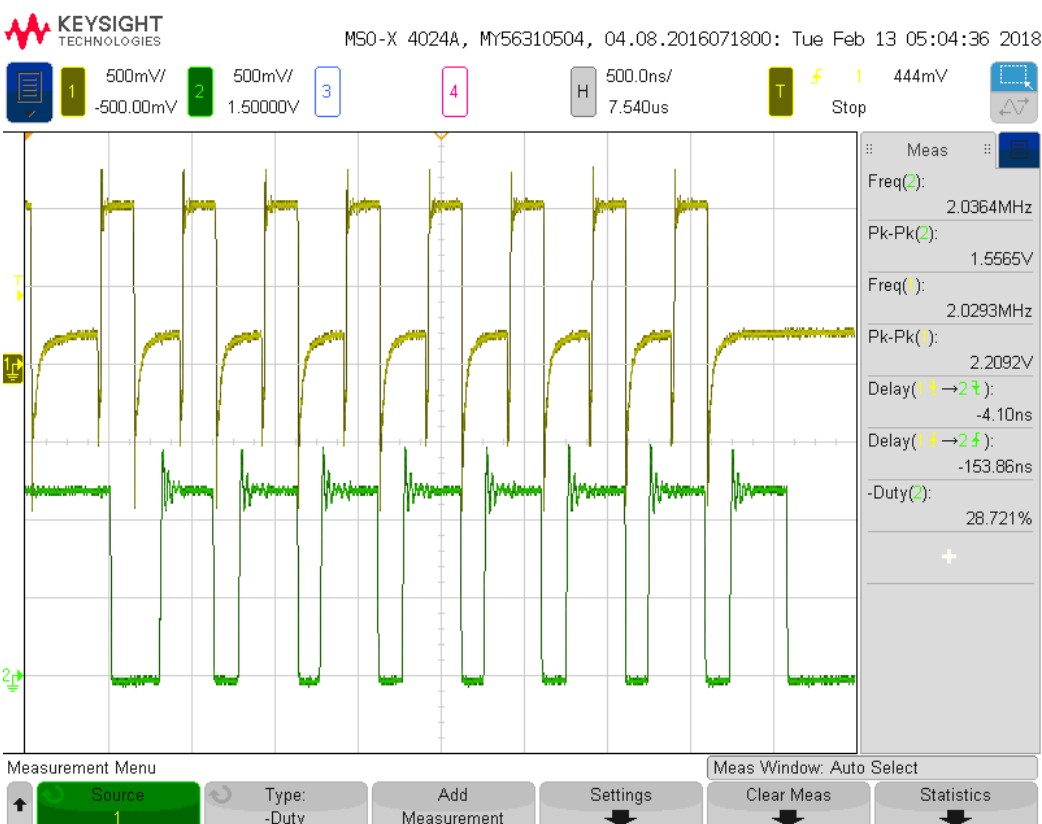

**Figure 4.** First system configuration measurement (no load at 4.07 MHz).

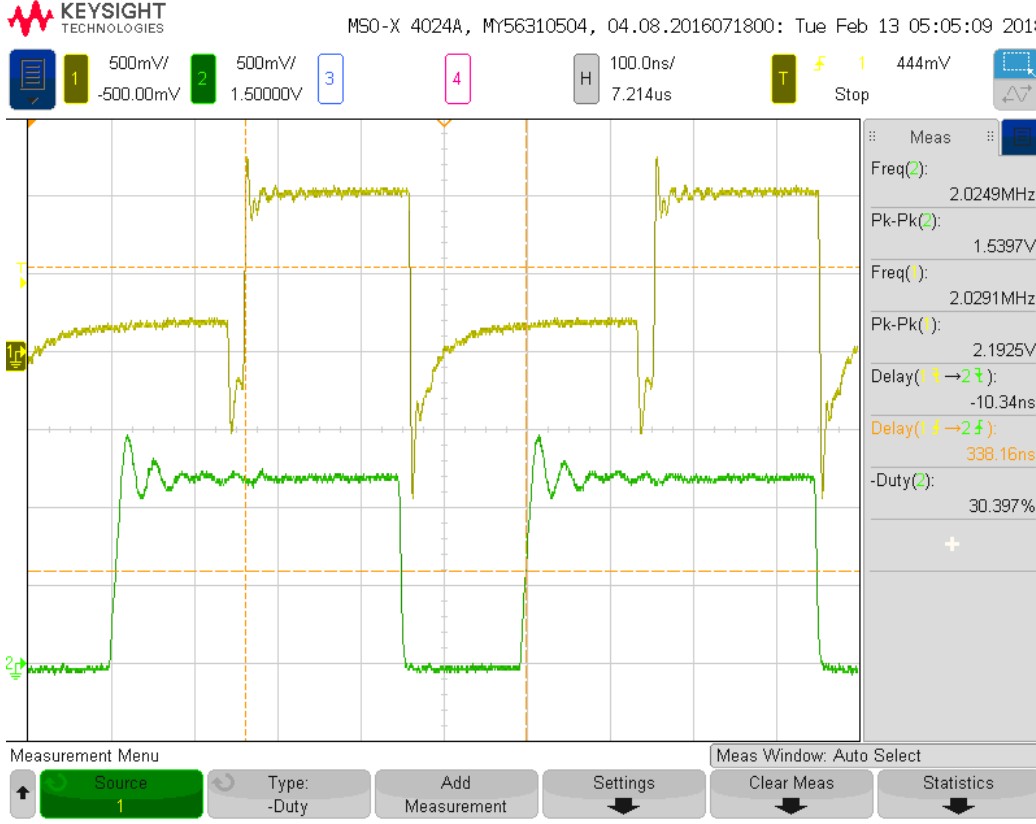

**Figure 5.** First system configuration measurement for rising edge propagation delay.

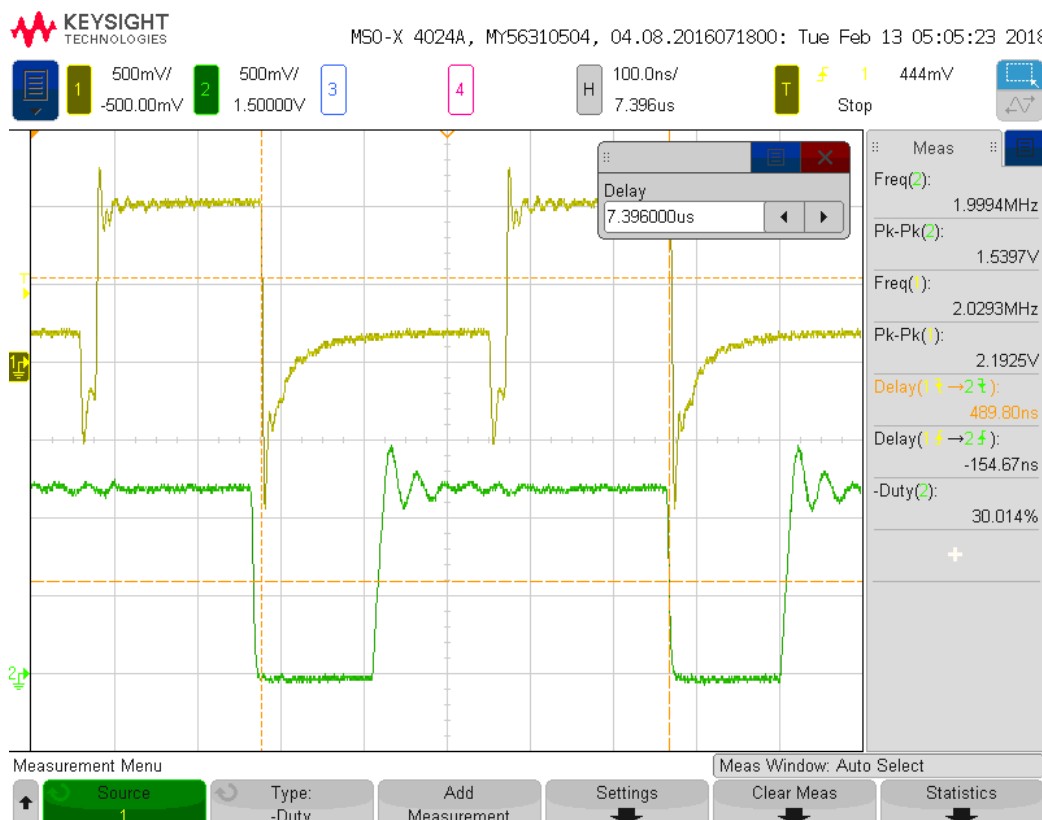

**Figure 6.** First system configuration measurement for falling edge propagation delay.

Figure 7 depicts the seventh configuration analyzed for measurement results with 4 inputs and 12 neurons. The seventh configuration was chosen to detail the full utilization of all input signal pathways and all available resources in the programmable architecture. Figure 8 details three of the input voltage waveforms in yellow, blue, and magenta (oscilloscope was limited to four probes) with the output voltage waveform in green for the seventh configuration. The frequency for the input signal data is 1.51 MHz, which is reflected in the output waveform that goes from a low state at zero to a high state at 1.2 V. The measured on-state current for this configuration is 35.8 µA with the off-state current being 17 µA. These currents averaged over the 8 on states and 8 off states of the signal and multiplied with the voltage give an average power of 31.68 µW. This configuration has 60 total operations (5 per neuron with 12 neurons). Therefore, the power efficiency is 2.86 TOPS/W for this configuration and input signals. Figures 9 and 10 show the propagation delay for the rising and falling edges, respectively. The rising edge propagation delay is 555 ns, while the falling edge is 889 ns.

The remainder of the configuration figures are not shown so as not to overwhelm the result sections with excessive figures. However, their results are summarized in Table 1, showing the important configuration characteristics similar to the discussion in the previous paragraphs. Table 2 details each configuration's rising and falling edge propagation delay in nanoseconds. Table 3 details a comparison of this work against reviewed literature. The table is broken into several comparing features of the different system configurations with the units for each feature being for the computational density operations per second per µm$^2$ (Ops/s/µm$^2$), power in microwatts (µW), synapses (number of multipliers), density per synapse (Ops/s/µm$^2$), power per synapse (µW), and power efficiency (TOPS/W). Table 4 shows each design's implemented technology node or whether software only, the nominal supply voltage, and nominal frequency as compared with the proposed architecture.

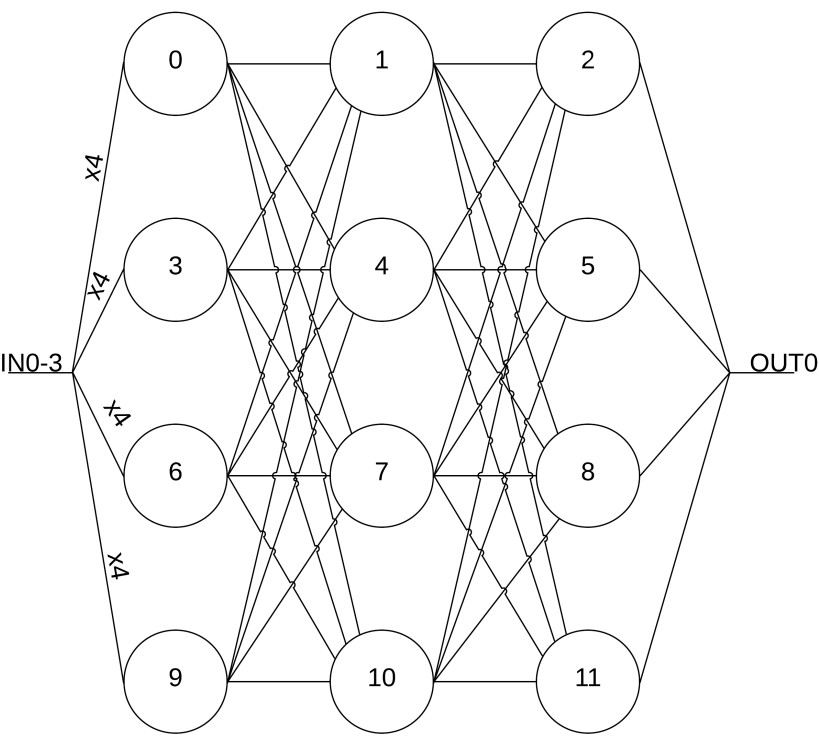

**Figure 7.** Hardware configuration for seventh configuration.

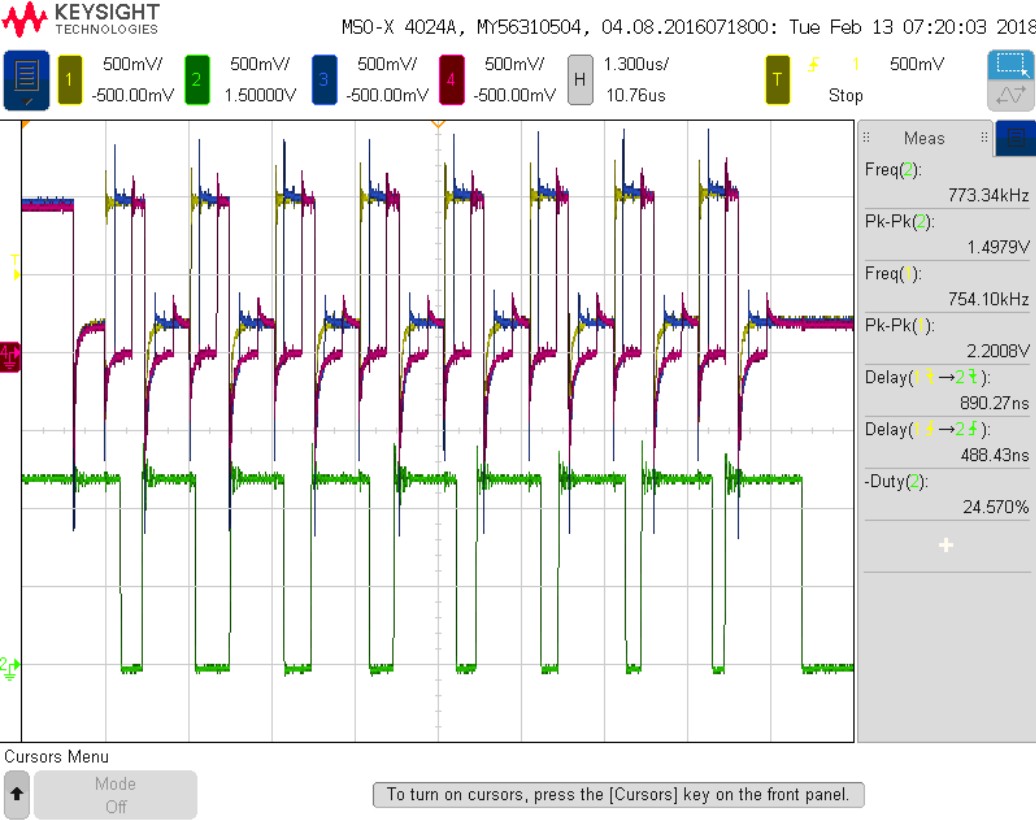

**Figure 8.** Seventh system configuration measurement (no load at 4.07 MHz).

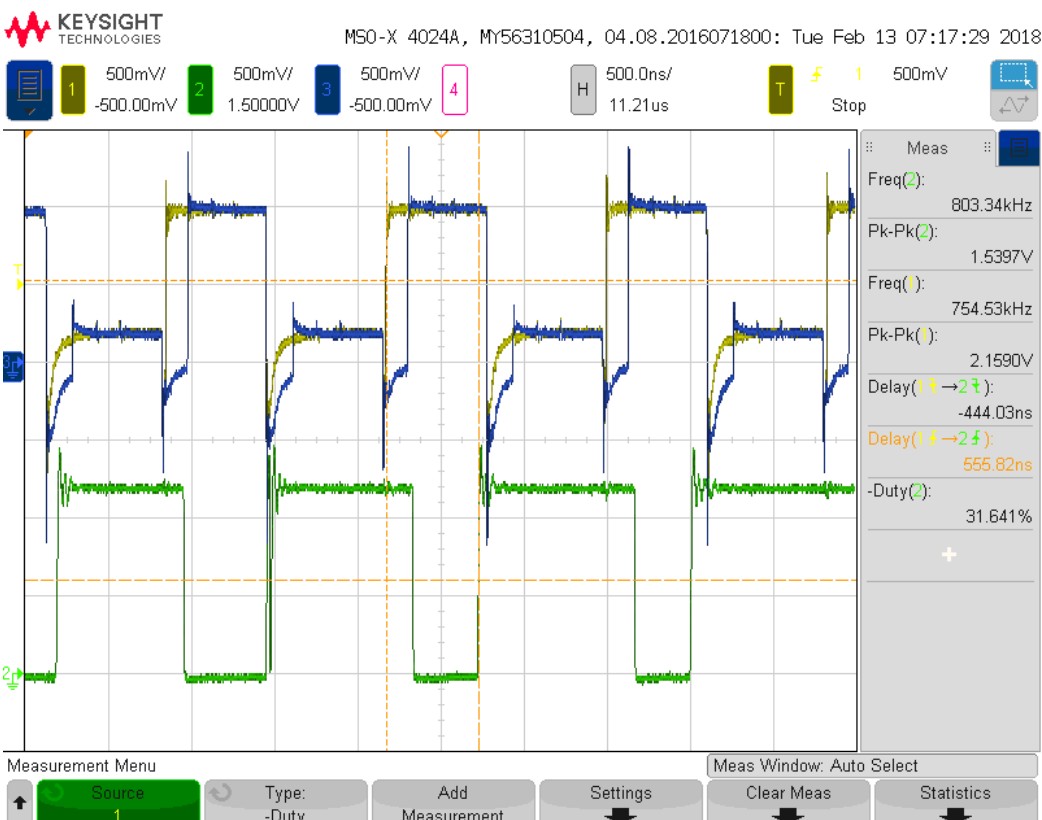

**Figure 9.** Seventh system configuration measurement for rising edge propagation delay.

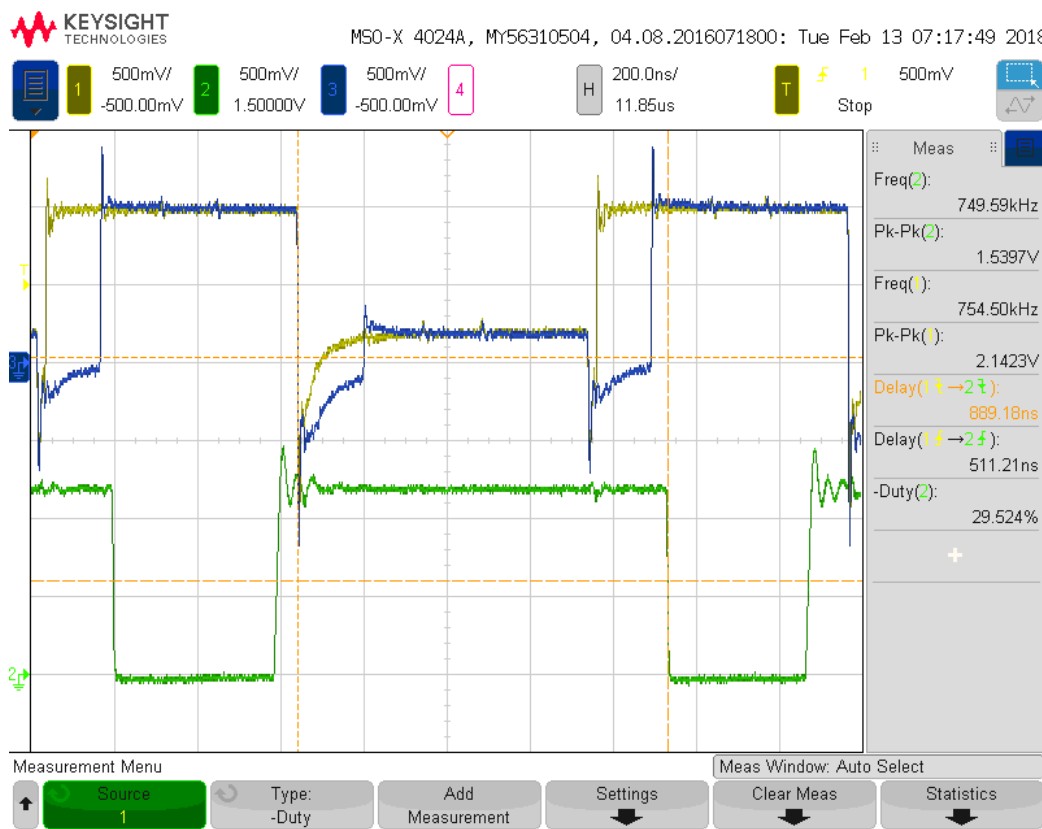

**Figure 10.** Seventh system configuration measurement for falling edge propagation delay.

**Table 1.** Summary of measurement results for different system configurations.

| Config Number | Inputs | Neurons | Freq. (MHz) | On/Off Current (µA) | Power (µW) | Power Efficiency (TOps/W) |
|---|---|---|---|---|---|---|
| 1 | 1 | 3 | 4.06 | 14/7 | 12.6 | 1.93 |
| 2 | 1 | 4 | 4.06 | 14.1/7.3 | 12.84 | 2.52 |
| 3 | 1 | 5 | 4.06 | 19.4/8.6 | 16.8 | 2.90 |
| 4 | 1 | 7 | 4.06 | 18.9/6.3 | 15.12 | 5.91 |
| 5 | 1 | 9 | 2.75 | 22.8/7.4 | 18.12 | 4.56 |
| 6 | 1 | 12 | 2.75 | 26.6/15.5 | 25.26 | 5.23 |
| 7 | 4 | 12 | 1.51 | 35.8/17 | 31.68 | 2.86 |
| 8 | 1 | 6 | 2.04 | 13.5/4 | 10.5 | 2.09 |
| 9 | 2 | 6 | 2.57 | 19/8.1 | 16.26 | 2.85 |
| 10 | 1 | 7 | 4.06 | 21.6/8.2 | 17.88 | 4.54 |

**Table 2.** Summary of propagation delays for different system configurations.

| Config Number | Inputs | Neurons | Freq. (MHz) | Delay (Rising, ns) | Delay (Falling, ns) |
|---|---|---|---|---|---|
| 1 | 1 | 3 | 4.06 | 338 | 489 |
| 2 | 1 | 4 | 4.06 | 345 | 457 |
| 3 | 1 | 5 | 4.06 | 343 | 479 |
| 4 | 1 | 7 | 4.06 | 436 | 492 |
| 5 | 1 | 9 | 2.75 | 515 | 585 |
| 6 | 1 | 12 | 2.75 | 541 | 621 |
| 7 | 4 | 12 | 1.51 | 555 | 889 |
| 8 | 1 | 6 | 2.04 | 829 | 750 |
| 9 | 2 | 6 | 2.57 | 610 | 687 |
| 10 | 1 | 7 | 4.06 | 395 | 564 |

**Table 3.** Hardware comparison with prior art.

| Prior Art | Analog or Digital | Power (µW) | Density per Synapse (Ops/s/µm$^2$) | Power per Synapse (µW) | Power Efficiency (TOps/W) | Fabricated |
|---|---|---|---|---|---|---|
| Park [9] | Digital | 213,100 | 20.00 | 103.65 | 1.930 | Yes |
| Tsai [10] | Digital | 310,000 | 23.86 | 75.68 | 1.450 | Yes |
| Yuzuguler [11] | Analog | - | 40.64 | - | 3.846 | No |
| Binas [12] | Analog | 200 | - | 0.008 | 7.97 | Yes |
| Hoang [13] | Digital | 42,100 | - | - | 0.0012 | No |
| Shreyas [14] | Digital | 10,450 | - | - | 2.60 | No |
| Zhang [15] | Digital | 7,250,000 | - | - | 0.0524 | No |
| This work | Analog | 15.12 | 8.12 | 2.16 | 5.91 | Yes |

**Table 4.** Technology, voltage, and frequency comparison with prior art.

| Prior Art | Technology (nm) | Supply Voltage (V) | Operating Frequency (MHz) |
|:---:|:---:|:---:|:---:|
| Park [9] | 65 | 1.2 | 200 |
| Tsai [10] | 65 | 1.2 | 210 |
| Yuzuguler [11] | 65 | - | - |
| Binas [12] | 180 | 1.8 | 0.066 |
| Hoang [13] | 65 | 1.2 | 50 |
| Shreyas [14] | 65 | 1/0.55 | 294/53 |
| Zhang [15] | Software | - | 200 |
| This work | 130 | 1.2 | >1 |

The noise analysis for the configuration in Figure 3 is the last system performance measurement. The noise is measured with the hardware set up as a simple classifier with two classes (OFF being class 0 and ON being class 1). An input is connected to a current source meter to provide the input current to measure input-referred noise. When the input is near the decision boundary for the system and the classification operation is performed, the noise will cause the output to become uncertain. Assuming additive Gaussian noise, the relative frequency of the class 1 output will be shown to approach the cumulative density function (CDF) of the normal distribution. The standard deviation $\sigma$ of this distribution can be extracted from the data and can be analyzed as the input-referred rms noise of the system. Two noise analysis runs are performed on this configuration and are shown in Figures 11 and 12. The measured input-referred noise for the first run is 435.9 $pA_{rms}$ with the second run giving a similar result of 432.13 $pA_{rms}$. With a full-scale input of 100 nA (or greater), the SNRs of the MLP system for this configuration for the first and second run are 47.21 and 47.29 dB, respectively. The analysis measures the circuit robustness within system configurations to develop larger neural network architectures utilizing the detailed subsystems.

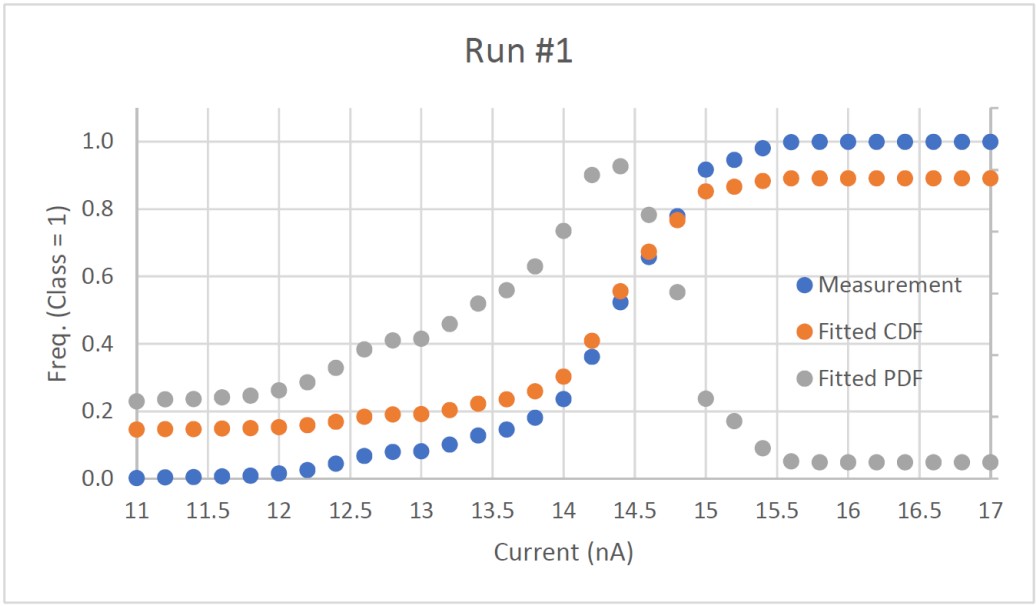

**Figure 11.** First noise analysis run on system.

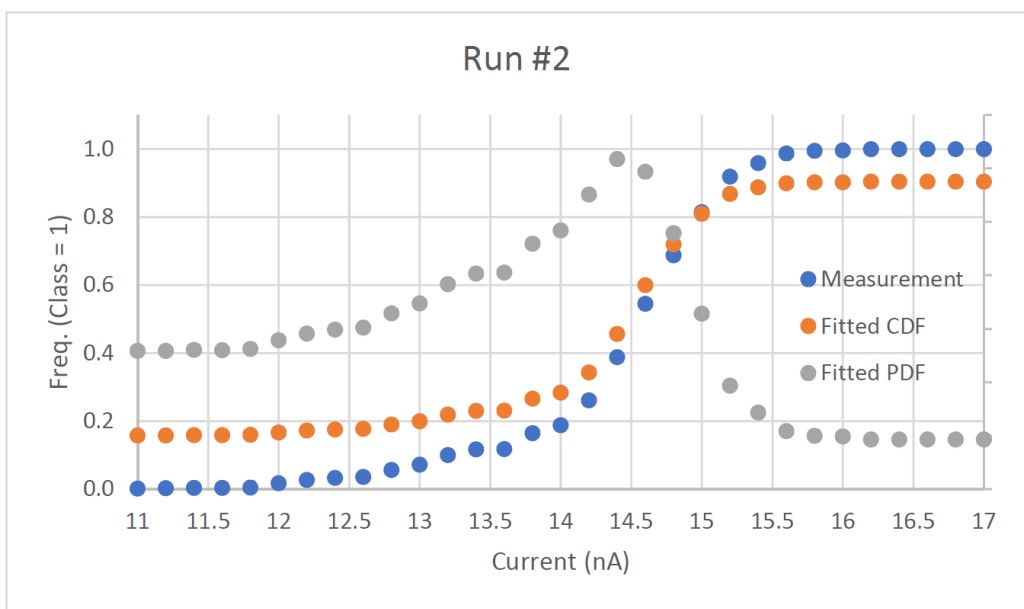

**Figure 12.** Second noise analysis run on system.

## 3. Materials and Methods

The programmable, energy-efficient analog MLP system contains neurons, winner-take-all (WTA) circuits, and switching structures as well as several support structures. The neurons are composed of several multipliers and one sigmoid circuit that are implemented via weak inversion operation, providing increased power efficiency while creating exponential characteristics within the circuits [16]. The support circuitry consists of numerous copies of biasing cells, switching cells, and shift registers that provide power, connectivity, and programmability, respectively. The system structure has been constructed with many copies of these subsystems to create an analog configurable neural network. The block diagram for the MLP is detailed in Figure 13 without the bias cells and shift registers. The system has four inputs that are routed to the neurons with two different types of switching matrices (detailed further in Section 3.4). The switching matrices offer the ability to create an MLP structure with 12 neurons in a fashion that the user requires. For example, the user could utilize a single neuron in the first layer that outputs to the next layer containing 4 neurons that in turn output to the next layer of 2 neurons that could then output to the winner-take-all circuitry before the signals exit the integrated circuit. The system limits each layer to four since each neuron only has four input multipliers, which is to simplify the overall structure to focus on the programmable and low-power system design. The main objective for this small MLP design is to merge the programmability capable within digital circuits with the energy efficiency achievable within analog circuits to implement a system that contains constructs that are scalable and improve the power efficiency of the simple neural network. Section 3.4 details the general operation of the overall architecture, while Sections 3.1–3.3 describe in further detail the circuitry that composes the system architecture.

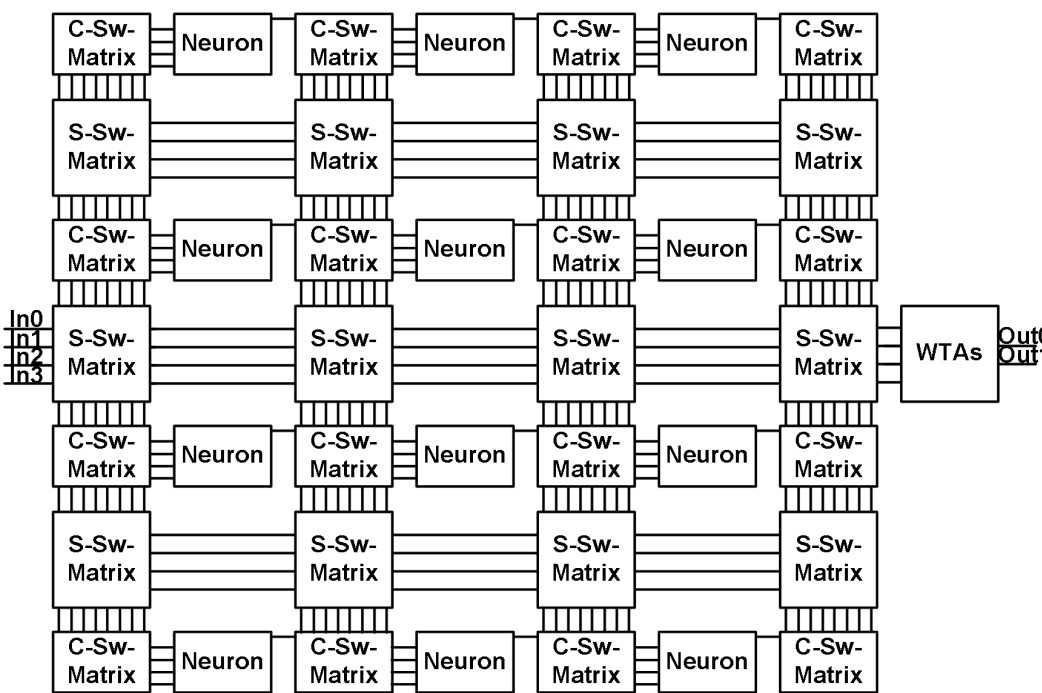

**Figure 13.** MLP hardware block diagram for programmable energy-efficient system.

### 3.1. Multiplier Design

The multiplier in this work is based on the translinear principle developed by Gilbert [17]. The basic translinear principle states that any closed loop containing an equal number of devices oriented in both directions (clockwise and counterclockwise loops) creates a circuit where the product of currents in one orientation equals the product of currents in the other orientation. Gilbert created the first translinear cells consisting of bipolar transistors, whereas today designers can utilize MOSFETs in weak inversion to create the same operation [18].

As the following equations will demonstrate, the multiplication and division operation of the Gilbert translinear cell is completely reliant upon four currents ($I_1$ through $I_4$) and not upon device parameters.

$$J_1 J_2 = J_3 J_4 \tag{3}$$

Equation (3) is based on the translinear principle equating two current loops of equal *pn* junctions where $J$ represents the current density of a bipolar transistor.

$$J_4 = J_1 \frac{J_2}{J_3} \tag{4}$$

Equation (4) is a manipulation of Equation (3) to isolate a single device's current density relative to that of the other three devices.

$$I_4 = I_1 \frac{I_2}{I_3} \tag{5}$$

$$A_1 A_2 = A_3 A_4 \tag{6}$$

Equation (5) represents removing the device sizing characteristics from the current and is only true when Equation (6) is true. Equation (6) states that the device sizing (in this case, the emitter area of the bipolar devices) of the first two transistors must equal that of the second two for the current multiplication or division in Equation (5) to be valid. Gilbert states in [19] that the devices need not all be the same size but rather only equal in the same amount of area when their area is multiplied with the area of their corresponding device. The Gilbert cell configuration coupled with weak inversion operation creates the

opportunity to develop a MOSFET circuit capable of operating with improved energy efficiency via current-mode operation in a neural network application.

With control of individual transistor characteristics, the multiplier circuit can be biased to operate in weak inversion at low current levels. This ability opens up the possibility of a low-power solution for the multiplier circuit. Weak inversion utilization offers exponential transfer characteristics and low power consumption [7]. The MOSFET weak inversion equation is compared with the bipolar exponential equation in the following two equations:

$$I_D = I_0 \frac{W}{L} \exp(\frac{\kappa V_G - V_S}{U_T}) \quad for \, V_{DS} > 4U_T \tag{7}$$

$$I_C = I_S \exp(\frac{V_{BE}}{U_T}) \tag{8}$$

Equation (7) depicts the MOSFET weak inversion drain current, where $I_0$ is a process-dependent constant, $W$ is the width of the transistor, $L$ is the length of the transistor, $\kappa$ is the gate coupling coefficient, $V_G$ is the gate voltage, $V_S$ is the source voltage, and $U_T$ is the thermal voltage [16]. Equation (8) represents the collector current in a bipolar transistor, where $I_S$ is the saturation current and $V_{BE}$ is the base-emitter voltage [20]. Equations (7) and (8) are highly similar to the largest difference coming from the gate coupling coefficient ($\kappa$) in the MOSFET equation. If assuming that $\kappa$ equals one, then the transconductance of a MOSFET and a bipolar transistor are the same, leading to a similar operation between the weak inversion MOSFET drain current and the bipolar collector current. Furthermore, Equation (7) directly changes Equation (3) to be the following for currents:

$$I_1^{\frac{1}{\kappa}} * I_2 = I_3^{\frac{1}{\kappa}} * I_4 \tag{9}$$

While the $\kappa$ term does not cancel out from both sides of Equation (9), the variance experienced by the inclusion of the $\kappa$ term is taken care of by the biasing/weighting constructs implemented within the system and does not appreciably affect the functionality of the multiplier within the overall system. While the MOSFET weak inversion operation varies greatly within real devices, these equations demonstrate the ability to utilize MOSFETs as exponential devices similar to BJTs for the implementation of the system circuitry.

The multiplier design consists of two different sections. The first section implements the multiplication operation through a circuit similar to Gilbert's bipolar cell. The second section implements two Minch cascode current mirrors to provide the bias or weight signals ($I_2$ and $I_3$ in Equation (5)). These signals scale the input signal according to the desired weight characteristic being implemented. Figures 14 and 15 show the circuit diagrams for the multiplier cell and the Minch cascode current mirrors, respectively. The dimensions of the transistors are not shown on the figures, but the minimum-sized devices are used for maximum area density within a neural network application (width of 360 nm and length of 240 nm). These devices are thick oxide transistors used to take advantage of their higher threshold voltage for better control of the weak inversion signals.

The multiplier utilizes a current-mode approach to decrease the propagation delay of the circuit from long wire traces on-chip. The current-mode operation along with the weak inversion biasing allows the circuit to operate at increased energy efficiency. Current-mode operation offers the capability for higher data rates due to lower signal propagation delay times due to the parasitics seen by the current signal. Lower-power operation is obtained by biasing the circuitry in the weak inversion region, which also maintains the exponential transistor characteristics required for successful multiplication. Figure 14 depicts the core circuitry that implements the multiplication operation and is similar to Gilbert's cell. The five core transistors exhibit a similar translinear loop seen in Gilbert's cell due to the exponential characteristics of the MOSFET weak inversion operation. The addition of the cascode device in the $I_1$ input pathway allows the multiplier to operate faster while maintaining signal integrity via a low impedance input node for the current input signals. Otherwise, there would be significant current division between the input node and the

source impedance, resulting in attenuation in the signal pathway. The circuit operation is as follows: the input signal enters via the $I_1$ pathway, the input signal is scaled with the $I_{2m}$ and $I_{3m}$ weight signals from the Minch cascode circuits, and $I_4$ sinks the current, producing a scaled version of the input signal with some variations due to transistor mismatch and weak inversion operation.

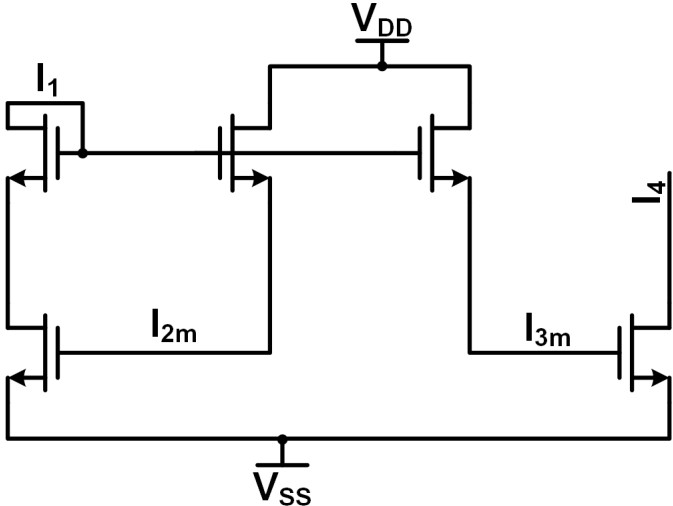

**Figure 14.** Multiplier cell schematic.

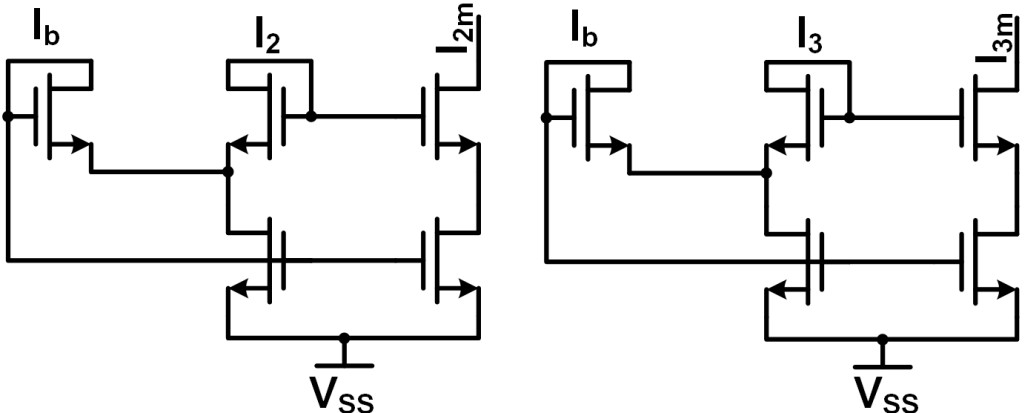

**Figure 15.** Minch cascode bias cell schematics.

The circuits in Figure 15 represent two Minch cascode biasing schemes that transfer the weight inputs to the multiplier for adjustment of the input signal. The Minch cascode circuit from [21] creates a low-voltage cascode structure that allows for weak inversion bias weights while maintaining a lower-voltage rail for the circuit. The Minch cascode circuit also provides a more accurate current mirroring operation than other simpler current mirror designs. This effect allows for the multiplier weights to be more reliably reproduced as they are programmed for physical circuits, even accounting for transistor mismatch and operation in the weak inversion region. The key to this increased accuracy comes from the input bias current signal ($I_b$) that helps reduce the weak inversion current mismatch [21]. The Minch cascode scheme allows the transistors to operate effectively during switching operations within the minimal voltage headroom seen in modern process technologies. Since the weight signals are DC values and do not require high-speed operation like the input signal $I_1$, the Minch cascode cells can easily be integrated with the multiplier cell while maintaining the weak inversion weights. Figure 16 details the simulation results for a 100 nA input signal into the multiplier when the weight and saturation currents are also

set to 100 nA. The current output signal follows the input signal with some variation due to the utilization of subthreshold signals, but this variation will be resolved at the system level via adjustment of weights further down the signal processing line (at either the sigmoid circuit or the next neuron). Additionally, the added current is beneficial to maintaining the desired current signal levels while the signal propagates through the system. Figure 17 demonstrates some of the DC characteristics of the multiplier for set weighting signals of 100 nA each, while the input current is swept from 0 to 500 nA. The output signal at the bottom shows current variations due to the subthreshold design, but these variances will not impact the functionality of the following sigmoid circuit significantly and can be adjusted via the control of bias currents in either the sigmoid circuit or the multiplier circuit itself.

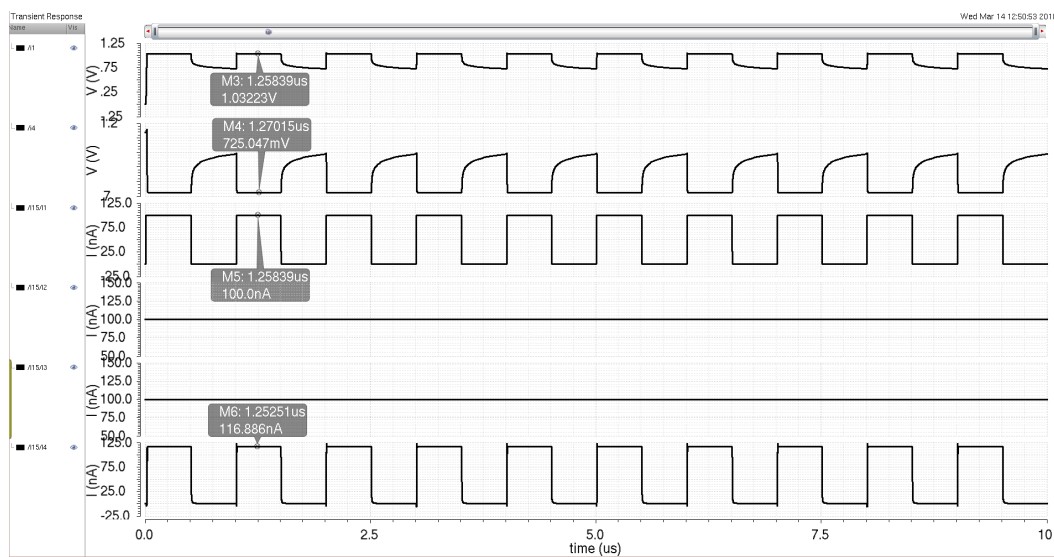

**Figure 16.** Simulation of the multiplier circuit with an input current of 100 nA and weighting signals of 100 nA.

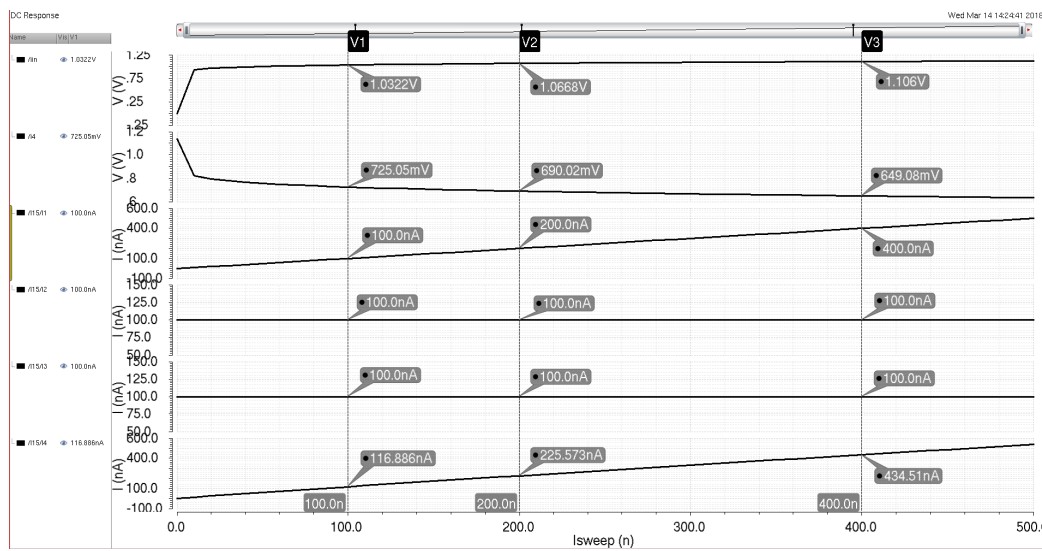

**Figure 17.** Simulation of the Multiplier Circuit with a DC sweep of the input current and weighting signals of 100 nA.

## 3.2. Sigmoid Design

The sigmoid circuit is an important base element in the MLP system, creating either a logistic function or hyperbolic tangent function output (Figure 18) at the end of the neuron

signal pathway that will propagate onto another neuron within the neural network. For this work, the logistic function is utilized since it does not require a negatively biased circuit to operate as in the hyperbolic tangent function. The utilization of current signals in the sigmoid circuit is desired to allow the MOSFETs to function in the weak inversion region while operating as quickly as possible.

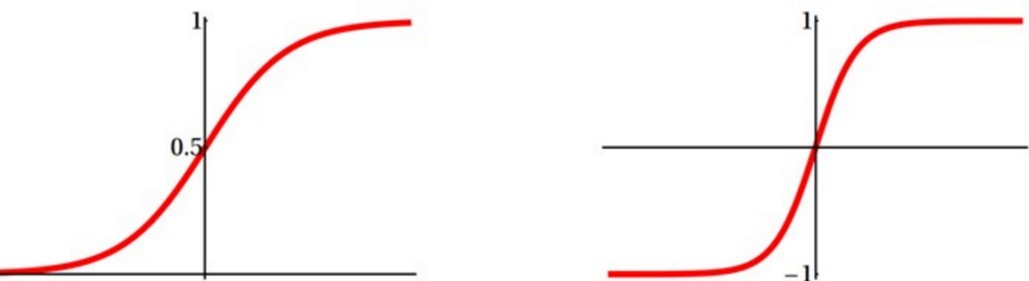

**Figure 18.** Logistic function (**left**) and hyperbolic tangent function (**right**).

The core circuitry for the sigmoid function consists of three transistors, a differential pair, and a tail (bias) current [22]. The differential output current of the MOSFET pair is quite similar to that of the bipolar pair with some differences that stem from using MOSFETs over bipolar transistors. Analyzing the circuit from Mead, the following equations are generated:

$$I_{sat} = I_0 \exp(V_g \kappa - V_s) \tag{10}$$

Applying the saturated drain current equation to the differential pair yields:

$$I_1 = I_0 \exp(V_1 \kappa - V) \tag{11}$$

$$I_2 = I_0 \exp(V_2 \kappa - V) \tag{12}$$

The drain currents added together must equal the tail bias current:

$$I_b = I_1 + I_2 = I_0 \exp(-V)(\exp(V_1 \kappa) + \exp(V_2 \kappa)) \tag{13}$$

Solving for $\exp(-V)$ and substituting into Equations (11) and (12) produces:

$$I_1 = I_b \frac{\exp(V_1 \kappa)}{\exp(V_1 \kappa) + \exp(V_2 \kappa)} \tag{14}$$

$$I_2 = I_b \frac{\exp(V_2 \kappa)}{\exp(V_2 \kappa) + \exp(V_1 \kappa)} \tag{15}$$

Taking the difference of Equations (14) and (15) gives the final tanh function:

$$I_1 - I_2 = I_b \frac{\exp(V_1 \kappa) - \exp(V_2 \kappa)}{\exp(V_1 \kappa) + \exp(V_2 \kappa)} = I_b \tanh \frac{\kappa(V_1 - V_2)}{2} \tag{16}$$

$$V_1 = V_{TH} + \sqrt{\frac{I_{in}}{\frac{W}{2L} * \mu_n * C_{ox}}} \tag{17}$$

$$V_2 = V_{TH} + \sqrt{\frac{I_{ref}}{\frac{W}{2L} * \mu_n * C_{ox}}} \tag{18}$$

$$I_1 - I_2 = I_b \tanh \frac{\kappa\left(\sqrt{\frac{I_{in}}{\frac{W}{2L} * \mu_n * C_{ox}}} - \sqrt{\frac{I_{ref}}{\frac{W}{2L} * \mu_n * C_{ox}}}\right)}{2} \tag{19}$$

For all of the above equations, $\kappa$ is a constant that represents the ratio of the MOSFET surface potential of the gate voltage [22]. The basis of these equations requires the differential pair to have voltage inputs. Since the multiplier design from the previous section outputs a current signal and the reference signal for the sigmoid circuit being a current signal, both need to be converted via diode-connected devices represented by Equations (17) and (18), leading to Equation (19), which is the output current difference based on the input current and reference current signals. Therefore, it has been established that a circuit including a differential pair will produce the desired sigmoid function if the input currents are transformed appropriately to voltages via the diode-connected devices, and the output currents are effectively subtracted from each other.

Figure 19 is the design obtained that accommodates high-frequency current signals and outputs the desired logistic function. The sigmoid design is composed of several current mirrors that relay the input, bias, and output signals to and from the differential transistor pair at the core of the circuit with all transistors having a minimum width and length of 360 and 240 nm, respectively. These width and length are the minimum size for the thicker gate oxide devices available within the 130 nm technology node that offer a higher threshold voltage, which in turn provides the capability for greater design margin within the subthreshold region of each device. The basic functionality of the circuit is that when the main input signal $I_p$ is below the reference input signal $I_n$, the output signal $I_{out}$ will be near zero current. When $I_p$ becomes much higher in magnitude than $I_n$, $I_{out}$ will then output a current that is near that of the bias current $I_b$ for the sigmoid. This behavior is typical of a logistic function at the positive and negative extremes on the horizontal axis. The reference current $I_n$ allows for the logistic function's half point to be shifted further up the horizontal axis (see Figure 18). This functionality requires the circuit to have more input current in order to reach the fully saturated bias current output. The functionality of the sigmoid circuit can only be obtained by operating in the weak inversion region, which enables the output to take the shape of the desired logistic function for proper signal propagation.

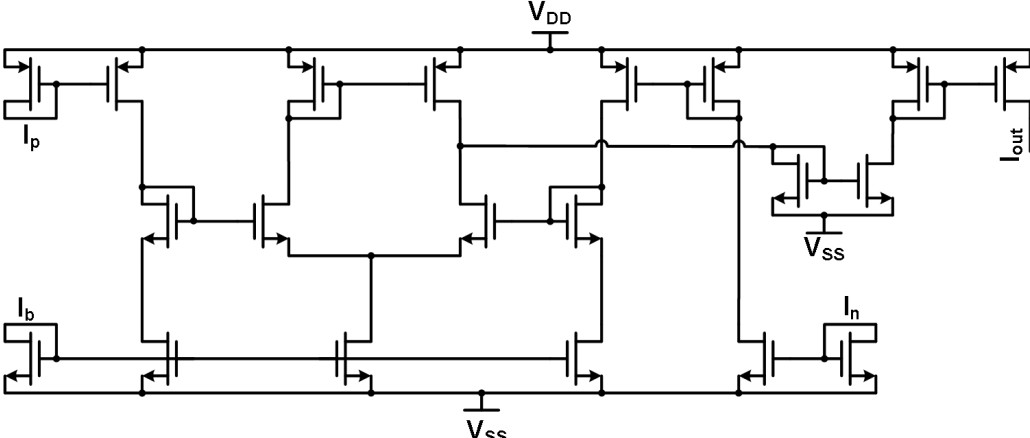

**Figure 19.** Sigmoid circuit schematic.

Figure 20 details the simulation results for the sigmoid circuit with a 100 nA switching current input signal and reference and biasing currents that produce an output signal that is similar in amplitude to the input signal. Chosen current values were based on the utilization of the bias cells (discussed in Section 3.4) with a power of 2 increments up and down from a system reference current of 100 nA. Therefore, a reference current of 25 nA was utilized alongside a bias current of 200 nA to produce an output current signal of around 93 nA for propagation to the next neuron in the system. Figure 21 shows the DC characteristics of the sigmoid circuit as the input signal is swept from 0 to 500 nA with a bias current of 200 nA and a reference current of 25 nA. The output current signal shows the activation function and the signal magnitude required for the logistic curve to start.

The subthreshold variance in the sigmoid circuit keeps the output signal from saturating at the bias current, but this variance can be dealt with in a similar fashion to the variance seen in the multiplier with the bias control system (adjustment of either the reference or bias currents to achieve a sufficient signal magnitude).

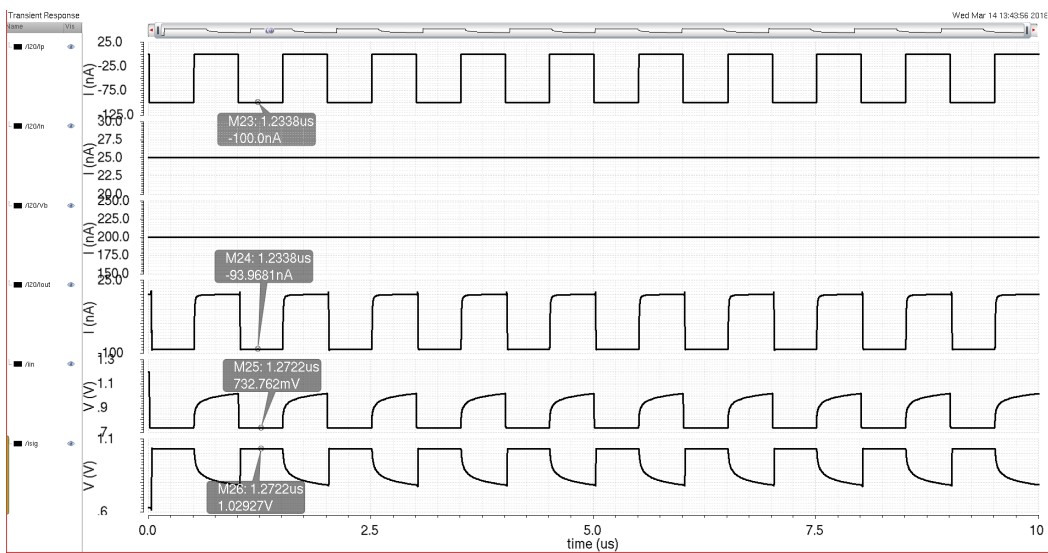

**Figure 20.** Sigmoid circuit schematic simulation with current and reference biasing to maintain a similar output current signal as the input signal.

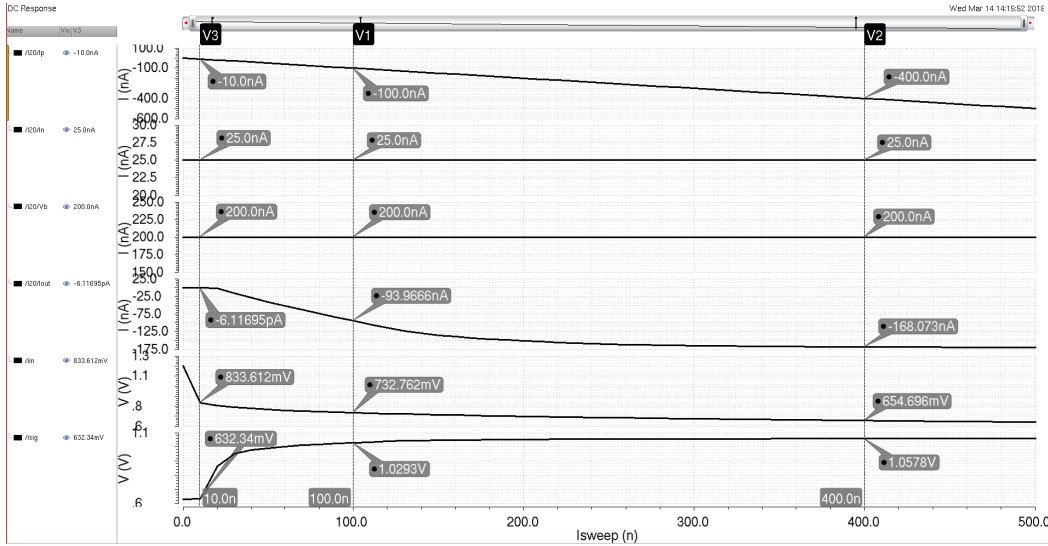

**Figure 21.** Sigmoid circuit schematic simulation with the input current swept from 0 to 500 nA with a constant reference and bias current of 200 and 25 nA, respectively.

### 3.3. Winner-Take-All Design

The final standalone circuit before the system circuitry is that of the thresholding circuit (TC) that creates a WTA circuit when combined with a 2-input OR gate. The TC receives neuron outputs and outputs a signal based on the highest signal level it receives at its input. Figure 22 shows a block diagram for the two WTA structures that correspond to the two system outputs. Each "complete" WTA design consists of a multiplier circuit (Figure 14) and a TC cell (Figure 23). The multiplier circuit sums the neuron current outputs at its input terminal and then scales the summed currents to create a better signal for comparison in the TC cell. The TC cell takes the multiplier output and compares it with a reference current level. This comparison determines if the signal should remain "high" or

"low" by creating a voltage at the comparison node that is either just above the threshold voltages of the following inverters or just below their threshold voltages. The use of the inverters provides the final output off-chip to be a digital voltage instead of an analog current signal that then needs to be converted. The inverters have different minimum widths (160 nm) and lengths (120 nm) compared with that of all the other transistors to provide faster functionality as they create the digital output voltage and utilize less physical space. The different minimum width and length stem from utilizing the standard gate oxide devices that are the main transistor devices available within the 130 nm technology node.

The TC cell in Figure 23 contains three current inputs and one voltage output. The main input signal $I_{in}$ that comes from the multiplier (and the neurons before that) goes into a PMOS Minch cascode current mirror to maintain the signal level and integrity before the comparison node. The reference current $I_b$ for current comparison is input into an NMOS Minch cascode current mirror similar to those in the multiplier circuit mentioned previously in this chapter. The Minch cascode structure requires a bias current $I_{b1}$ to operate effectively. The reference and bias current inputs are taken from system circuits and are DC values. The inputs $I_{in}$ and $I_b$ are mirrored and compared against each other at the comparison node before the two inverters. As mentioned before, this node fluctuates based on the input signal $I_{in}$ around the inverter threshold voltages. The output signal $V_{out}$ is a digital voltage signal that is then passed on to an OR2 gate. The OR2 gate provides another level of comparison with another signal chain, ensuring that the MLP system output follows the winner-take-all concept. Figure 24 demonstrates the winner-take-all circuit functioning in a simulation environment. The input signal to one of the multipliers is a 100 nA switching current signal. The weight and saturation currents for the multiplier are both 100 nA with the biasing signal for the TC cell being the same level. These current signals lead to the winner-take-all circuit, producing a high digital output signal when the current input signal is high.

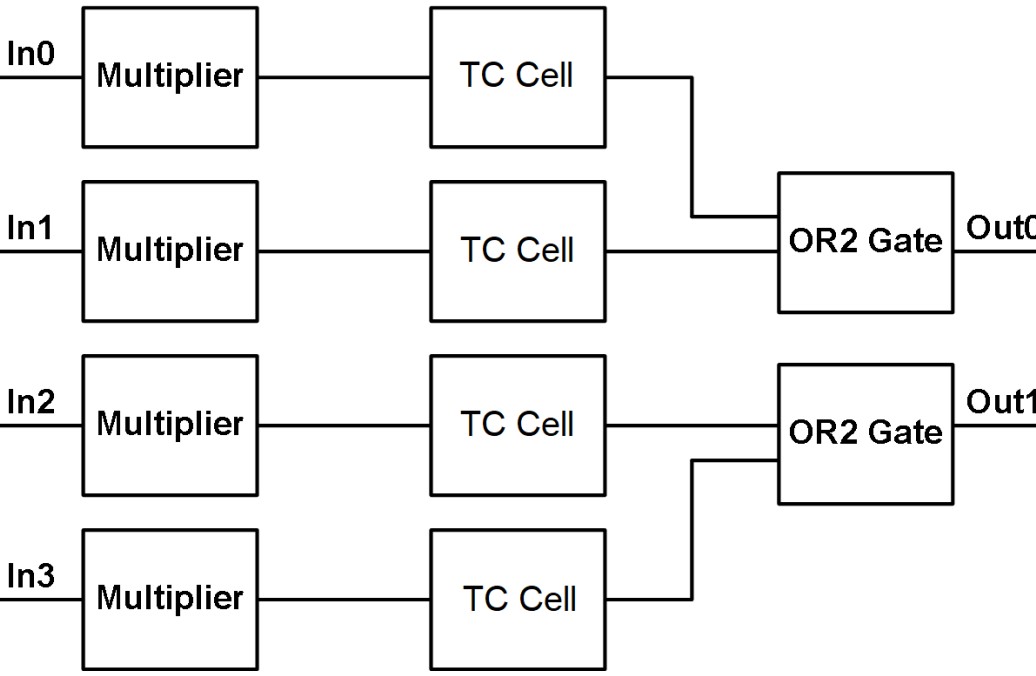

**Figure 22.** Winner-take-all block diagram.

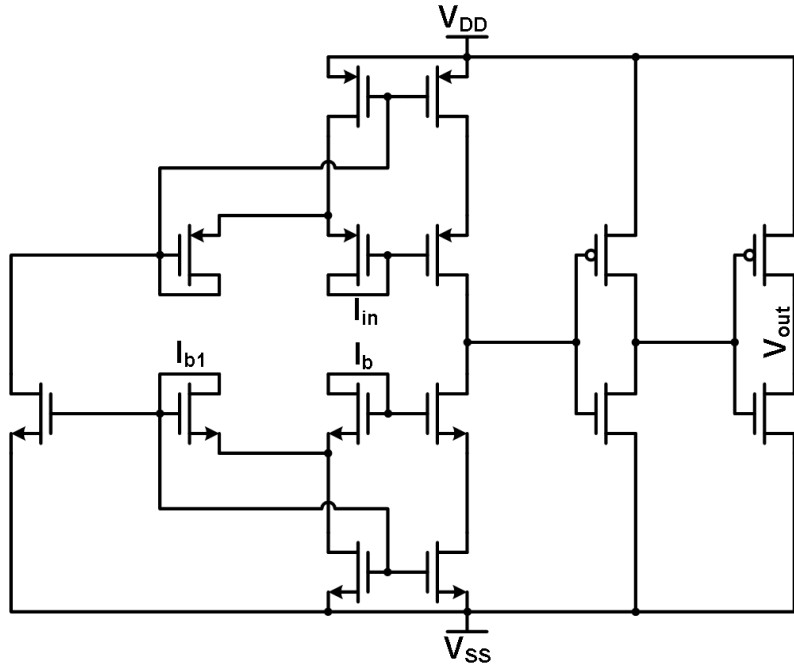

**Figure 23.** Thresholding circuit cell schematic.

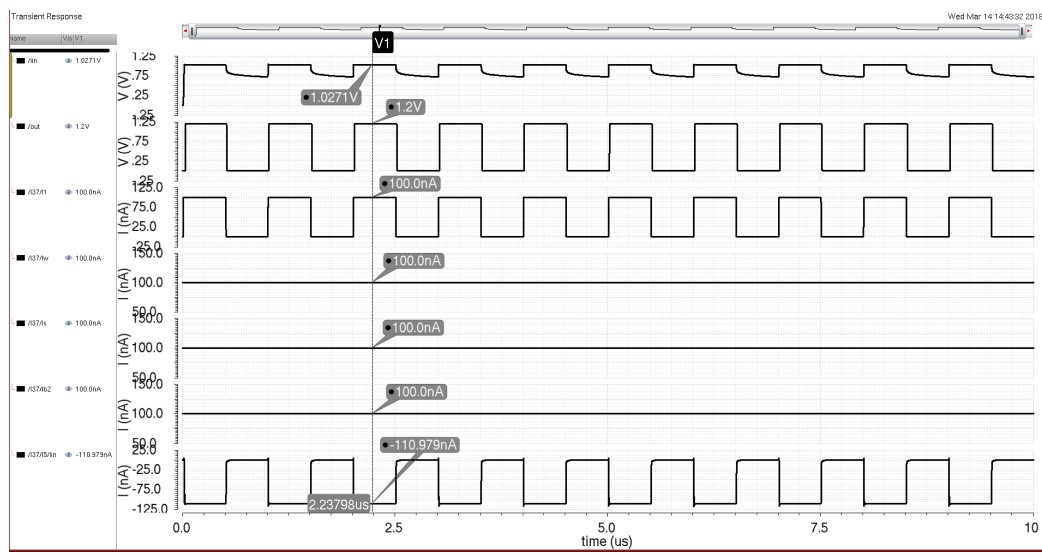

**Figure 24.** Winner-take-all circuit simulation with a 100 nA input current signal and 100 nA weight, saturation, and biasing signals to produce an output voltage that corresponds to the high signal levels of the input signal.

### 3.4. MLP Hardware System Architecture

The system design is based on the structure of field-programmable analog arrays (FPAAs) or FPGAs such that a similar programmable construct can be achieved. The architecture contains several hundred biasing cells that need to be programmed for the proper operation of the neural network. The basic biasing scheme flows as follows: first, the master bias current is sent on-chip; next, the master bias is mirrored to the bias control circuitry that controls whether currents are sent to the neuron/WTA blocks; lastly, the bias current is sent and mirrored into the bias cells based on how many of the current mirrors are programmed in each neuron/WTA block. The bias programming is controlled by a string of shift registers that are made up of basic D-type flip-flops, where an example of a 4-bit shift register can be seen in Figure 25. Figure 26 shows the master bias input current

structure and one of the current mirrors that would be controlled to send current to a single neuron or WTA block. Figure 27 details the bias input structure for the neuron and a single current mirror for the biasing of the neuron or WTA block. The programming of the neuron or WTA biasing determines the number of current mirrors in parallel for each bias current input.

The weighting of the multiplier and sigmoid circuits relies upon the programming of hundreds of bias cells similar to Figures 26 and 27 via shift registers connected directly to the "Sw" node between the transistors that set the current sink or source and VSS and VDD, respectively. Several bias cells have been put in parallel for each required bias current in the multiplier and sigmoid circuits to provide the weights required for an appropriate system operation. The bias cells for a single current signal would range from 0.25× to 8× or 16×, the master bias input current, which is 100 nA supplied externally. This methodology offers discrete current steps from 25 nA up to greater than 800 or 1600 nA, depending upon the circuit's needed biasing levels. These digitized current levels additionally provided programming for the user and increased design flexibility.

The system in Figure 13 consists of 12 neurons, 2 WTA blocks for the main system outputs, S-switch matrices, and C-switch matrices. Starting with the neurons, each contains four multipliers and a single sigmoid, which is shown in Figure 28. Normally, a neuron for a configurable architecture would have a much larger number of multipliers per neuron as each neuron should be capable of receiving inputs from every other neuron in the previous layer. The total inputs are limited to four per neuron due to IC process limits with metal layers and interconnects. The constraint on the inputs maintains signal integrity by decreasing routing and switching characteristics that would be needed for a higher number of multipliers, and helps constrain the required chip area for the hardware implementation.

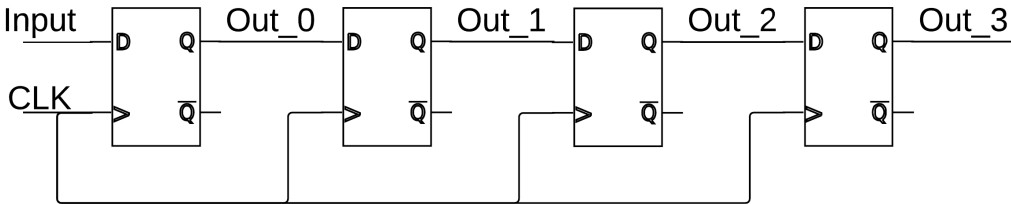

**Figure 25.** Four-bit shift register made up of D-type flip-flops.

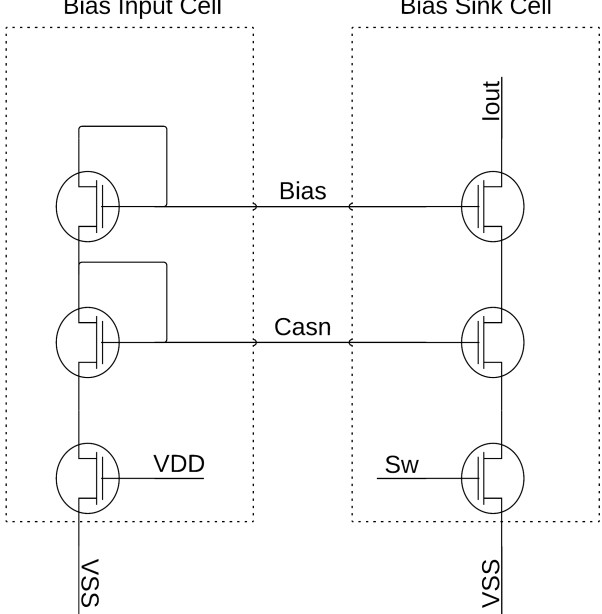

**Figure 26.** Master bias input cell with single current mirror output.

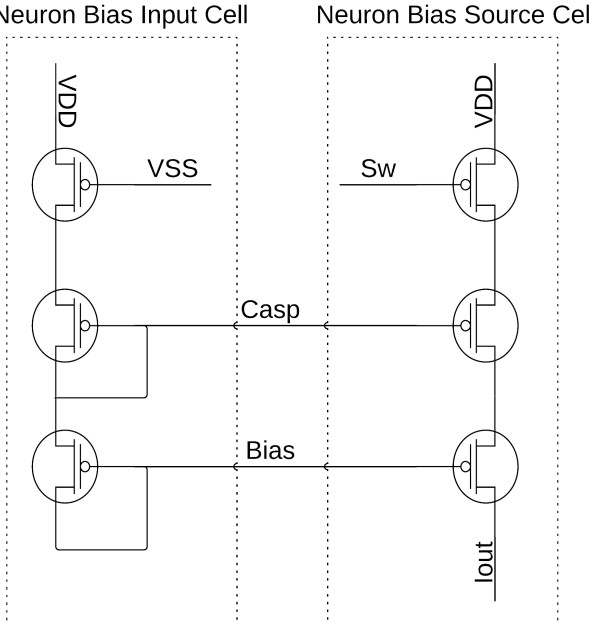

**Figure 27.** Neuron/WTA input bias cell with single current mirror output.

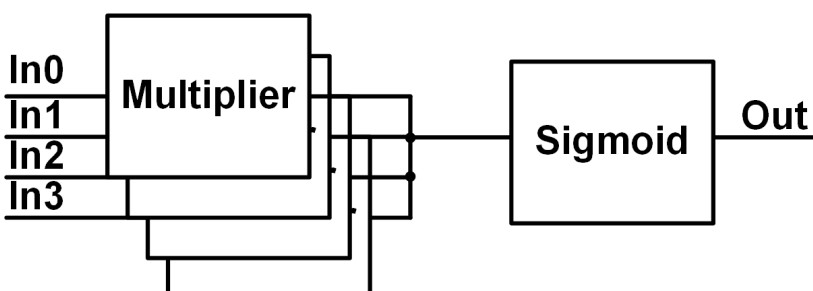

**Figure 28.** Neuron block diagram.

The next structures that will be discussed are the two switch matrices. The C-switch and S-switch matrices are developed from the switch matrix topology in [23] and take the forms shown in Figure 29a,b, respectively. The main difference is that the switches in the matrices are a single PMOS transistor whose gate is controlled by the output of a shift register instead of a floating gate node. The use of a single PMOS transistor instead of a floating gate cell for each switching node for connectivity reduces the complexity of the switching network and the programming complexity. The basic C-switch matrix has eight vertical routes that can be connected to five horizontal routes (four for neuron inputs and one for the output). Figure 30 shows these vertical and horizontal routes with the C-switch (single transistor) linking them together when activated. The S-switch matrix provides the ability to route a signal north, south, east, or west with as few switches as possible and can be seen in Figure 31. Each S-switch construct contains six transistors for the desired signal connections and a 3-bit address decoder to simplify the number of shift registers required to program a single S-switch structure. Additionally, the S-switch matrix offers routing through a layer of neurons to the next layer if desired. The connectivity is also stored within blocks of shift registers, providing a single desired configuration prior to the application of any input signals.

The system operates by routing one or more of the input signals to the first layer of neurons. Next, those neurons will weigh their inputs at the multipliers, sum the multiplier currents together before the sigmoid input, and then output a signal to the next layer of neurons. This operation will continue until the desired number of layers is achieved or all resources on the chip are utilized. The last layer of neurons will output to one or both of

the WTAs that will then amplify and compare the signal with a reference before outputting the digital version of the final signal. The system requires four data streams for switch programming, four data streams for neuron and WTA bias programming, and one data stream for master bias control programming.

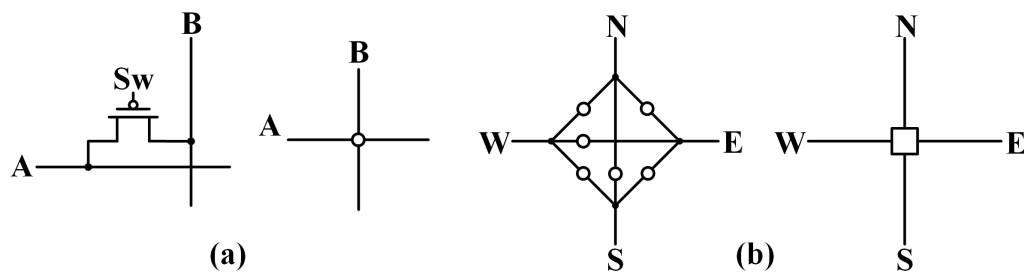

**Figure 29.** Switching structures: (**a**) C-switch and (**b**) S-switch.

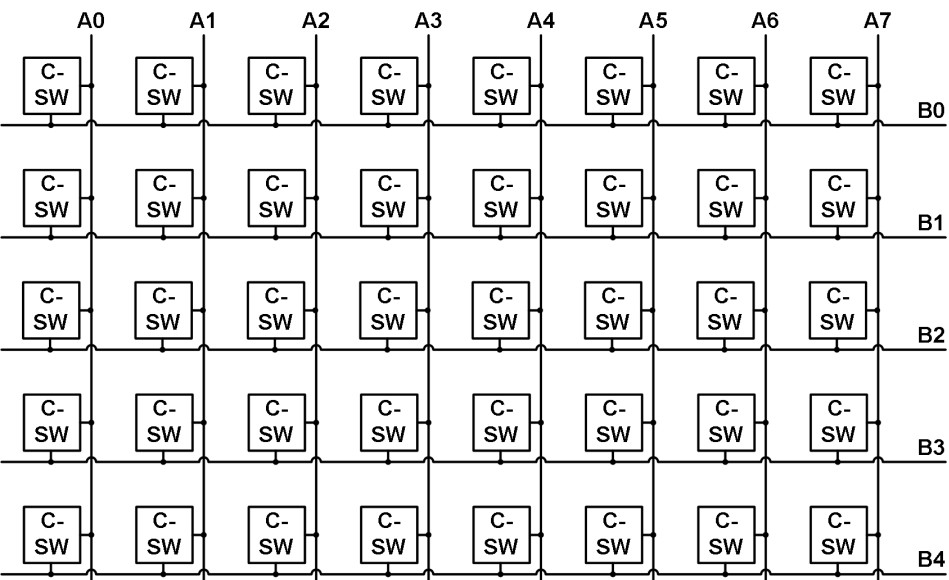

**Figure 30.** C-switch matrix.

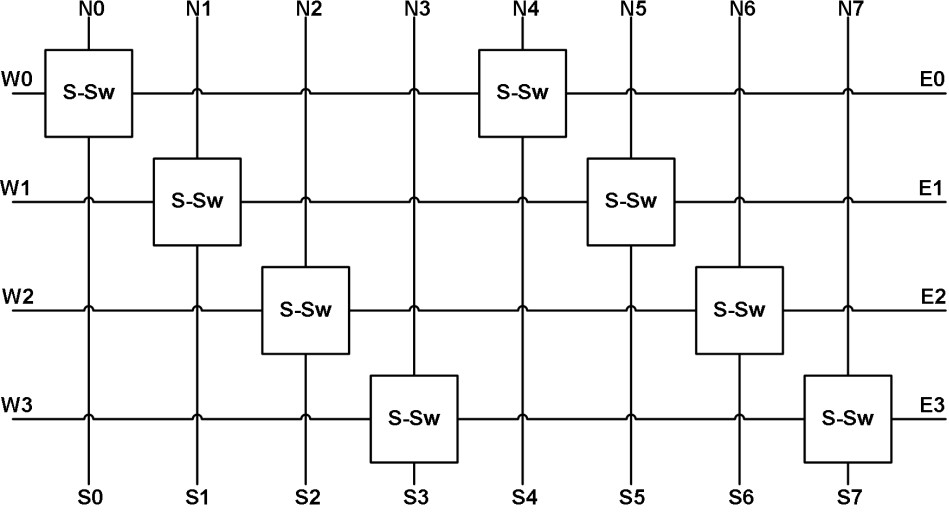

**Figure 31.** S-switch matrix.

The system implementation offers increased configurability for analog neural network hardware. The improved configurability stems from implementing an FPGA/FPAA-type routing scheme that allows signal pathways to be created going back and forth throughout the system to reach the desired resources. Additionally, the simplistic circuit structures utilized promote the scalability of the overall architecture by offering a "plug and play" type of circuit blocks that can readily be placed in a system architecture with little adjustment required to fine-tune the circuit performance. The biasing cells included within each neuron also provide greater flexibility to control multiplier weighting, sigmoid reference and biasing, and thresholding circuit response time. In this manner, the designer has the ability to control the power consumption and speed of the overall architecture via the programming of the neural network hardware. The programmability of the system offers the user the ability to create a diverse range of very basic neural networks that are capable of improved energy efficiency. The programmability also offers the capability to circumvent the effects that the subthreshold circuit variations have on the signal pathways. For example, the signal variations caused by the subthreshold design attribute to generally higher currents on the "high" levels of the signals. This variation can be neutralized by either adjusting the sigmoid circuit's reference or biasing levels or altering the biasing levels of other circuits downstream to be slightly lower to attest for the higher input levels. In this fashion, the analog variations that occur can be readily adjusted with little to no system level effects. Figure 32 shows the final layout view of the MLP programmable architecture that was sent for fabrication within the 130 nm technology node. The design with all circuit constructs was constrained to a 1 mm by 1 mm square layout space.

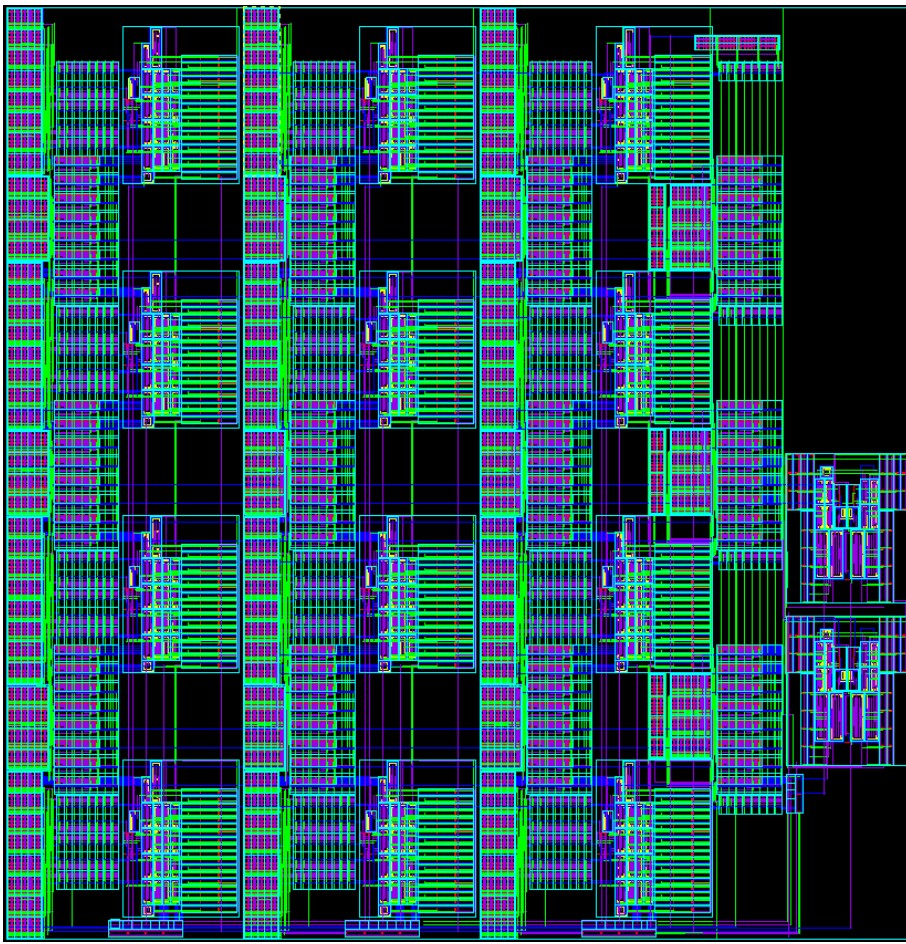

**Figure 32.** Final layout view of the MLP architecture that fits within a 1 mm by 1 mm physical space.

## 4. Discussion

The presented programmable and energy-efficient simple neural network design shows promising results compared with prior art in Tables 3 and 4. The system does not achieve the highest computational density or power per synapse but does boast a respectable power consumption and power efficiency compared with other works. The computational density is lower than comparable works due to the constraints placed on the neurons to provide the programmability (i.e., limited to four inputs) and the inclusion of support circuitry to offer greater designer control over power and signal propagation. Table 3 also demonstrates that an analog design that is not specifically implemented for a specialized function can outperform digital designs. However, the lack of system resources in this architecture does not warrant a direct comparison on typical neural network characteristics in terms of training and inference capabilities, but the small form factor designs for the multipliers, sigmoid, and WTA circuits coupled with the increased energy efficiency characteristics of analog, weak inversion design offers the ability to markedly scale up the neural network architecture in future works for a specialized functionality. The overall goal of achieving an analog hardware implementation of a programmable neuron array is to provide (1) scalability with a minimal number of transistors per neuron, (2) higher frequency operation bridging the gap between analog and digital designs, and (3) lower power consumption for pushing neural network architectures closer to edge capability.

## 5. Conclusions

The hardware system is centered around the ability to be both a programmable and energy-efficient analog network as analog multipliers and sigmoid circuits are already proven concepts. The first criterion of programmability is successfully demonstrated by the system via the ten configuration samples that produce the correct, expected outputs. The energy efficiency criterion is shown through the application and calculation of a power efficiency that is greater than 1 tera-operations per watt in each configuration. The system programming and small form factor allow for future designs to build up the number of available resources within the neural network that would enable a wide range of applications from image analysis to signal processing to pattern recognition. The architecture is well suited for this diverse range of applications if it is scaled up to become more of a true neural network as, right now, it is limited by connections and the number of inputs it can sustain. Classification measurements are not performed as circuit operation and functionality are the main focuses for this system prototype.

**Author Contributions:** Conceptualization, J.D. and J.H.; methodology, J.D.; software, J.D.; validation, J.D.; formal analysis, J.D.; investigation, J.D. and B.J.B.; resources, B.J.B.; data curation, J.D.; writing—original draft preparation, J.D.; writing—review and editing, B.J.B.; visualization, J.D.; supervision, B.J.B. and J.H.; project administration, B.J.B. and J.H.; funding acquisition, B.J.B. All authors have read and agreed to the published version of the manuscript.

**Funding:** This research received no external funding.

**Data Availability Statement:** No new data were created or analyzed in this study. Data sharing is not applicable to this article.

**Acknowledgments:** The authors would like to thank the University of Tennessee at Knoxville for their support and facilities in testing this work, the Integrated Circuits and Systems Lab at the University of Tennessee, and colleagues for their continual feedback and support during the design and testing of this work.

**Conflicts of Interest:** The authors declare no conflict of interest.

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
