# Peer review of "Programmable Energy-Efficient Analog Multilayer Perceptron Architecture Suitable for Future Expansion to Hardware Accelerators"

_jlpea, doi:10.3390/jlpea13030047_

Round 1
Reviewer 1 Report
This manuscript reports the programmable energy efficient analog multilayer perceptron architecture suitable for future expansion to hardware accelerators. This is a very good paper that can be accepted and published after addressing the following issues.
(1) It is suggested to remake Figure 1 to make it easier for readers to understand.
(2) Figure 2 requires a more detailed explanation.
(3) Some grammar error and spelling mistakes need careful correction by the author.
Good.
Reviewer 2 Report
The paper is well written and analyzed. My minor question is as follows.
Abstract mentions that the configurable system is fabricated in a standard 130 nm CMOS process. However, in page 7 line 193, the paper mentions that all transistors having a minimum width and length of 360 nm and 240 nm, respectively. Why the transistor length of 240 nm is used to implement the circuits (e.g., sigmoid circuit) instead of 130 nm? I think the minimum channel length should be 130 nm in 130 nm CMOS process. In page 8 line 219, the paper also mentions that the inverters have different minimum widths (160 nm) and lengths (120 nm). Authors should revise it properly.
Reviewer 3 Report
the paper is a complete engineering study of well known techniques. I cannot identify any new research result, neither at the circuit level nor at the system level.
Also, the claimed programmability has not been proven adequately.
Reviewer 4 Report
The authors have presented the design and implementation of a programmable analog multi-layer perceptron architecture which provides improvements in energy-efficiency compared to conventional digital designs. This is currently an important direction of research and the authors have proposed some innovative circuits and provided interesting experimental results. However, it needs to be revised to address the following concerns:
- In Section 3, please clearly mention which technology was used to fabricate the test chip. Include a chip micrograph and mention the core area required by the proposed MLP array along with nominal supply voltage and frequency.
- In Section 3, please provide a photo of the complete experimental setup. Currently, a photo of only the test board is provided.
- In Section 2, please provide simulation results for the individual circuit building blocks. Currently, only theoretical analysis and equations are provided. Please include simulation results to support that the circuits indeed behave as claimed.
- One of the biggest challenges in analog computation is the effect of transistor variability. How does the proposed architecture handle variability? What is the impact of transistor variations on the circuit accuracy? Please include Monte-Carlo or similar analysis.
- Since the proposed chip accelerates MLP computations, experimental demonstration of some neural network computation using the chip is also expected for completeness. What is the accuracy achieved for such an application compared to software or digital hardware implementation?
- There are many digital and analog machine learning hardware accelerators proposed in recent literature. Please expand Table 3 to include more such designs for better comparison. Table 3 must also include columns for technology node, supply voltage and operating frequency for fair comparison of the designs.
N/A
Reviewer 5 Report
In this manuscript the authors describe a reconfigurable architecture for implementing multilayer perceptrons using sub-threshold CMOS circuits. They provide some experimental results from a 12-neuron array fabricated in a 130-nm CMOS technology and some comparisons of the power efficiency of their work with other analog and digital approaches that have been published in the literature. In this reviewer's opinion, the work seems to be worthwhile, but the description of it in the present manuscript will require significant revision before it would be suitable for publication. In addition to improving the writing and some minor stylistic issues, the manuscript has some technical issues and leaves some important questions about the system unanswered. Two important questions that the manuscript leaves unanswered are: 1) How are the weights stored? and 2) How is the connectivity stored? On p. 11 in the paragraph between Fig. 12 and Fig. 13, the authors suggest that their C-switch and S-switch matrixes are "developed from floating gate switch cells in [16]," but it is not at all clear whether they are actually using floating-gate transistors as switches or if they are simply using a switch matrix topology similar to the one used in [16], perhaps with the state of the switches stored in the shift registers that they describe mention in other connections. Also, they do not seem to discuss weight storage at all. Are they supplying weights as external inputs at this stage of their work? Are they using some kind of floating-gate current memory cells? Are they using shift registers and some kind of current-output DAC? These seem like pretty major details of their system that the manuscript does not appear to address. In Section 2.1, the authors describe a translinear multiplier circuit that they use as synapses. At the bottom of p. 3, they write "Gilbert created the first translinear cells consisting of bipolar transistors whereas today designers can utilize MOSFETs in weak inversion to create the same operation [11]." Then, "...the multiplication and division operation of the translinear cell is completely reliant upon the four currents (I_1 through I_4) and not upon device parameters." Unfortunately, the authors seem unaware that Gilbert's classic translinear principle does not transfer over without qualification to sub-threshold CMOS. In fact, the four-transistor loop arrangement that they use as their synaptic multiplier, shown in Fig. 3, does *not* follow equation (2), which is I_1 I_2 = I_3 I_4 for devices of the same W/L ratio, but instead would follow I_1 I_2^\kappa = I_3^\kappa I_4, where \kappa is the device parameter that appears in their sub-threshold current model expressed in equation (6), which contradicts their claim cited above. [See. S.C. Liu, J. Kramer, G. Indiveri, T. Delbruck, and R. Douglas, Analog VLSI: Circuits and Principles, Cambridge, MA: The MIT Press, 2002, pp. 188-195 and Fig. 7.6a in particular.] Perhaps this mild dependence on a device parameter does not matter for the authors' application, the error in the presentation should be addressed prior to publication. In Section 2.2, the authors go into some detail analyzing the sub-threshold NMOS differential pair, showing that it follows a hyperbolic tangent relationship between its differential input voltage and its differential output current. There is an error in equation (14). In fact, it is I_1 - I_2 which is equal to the right-hand side rather than I_2 - I_1. More importantly, it is unclear why this analysis is even relevant given that they are not using the differential pair for their sigmoid unit in the manner that they have assumed for the analysis. They are using diode-connected NMOS transistors to convert input currents I_p and I_n into the input voltages to the differential pair. That would make the output current of the sigmoid circuit a bilinear rational function of the input currents rather than a hyperbolic tangent or a logistic function. If the authors are going to provide an analysis of the sigmoid, they should provide an analysis of the circuit shown in Fig. 6 rather than the one they have done on the differential pair. The results section is somewhat hard to interpret. The authors provide some screen shots of some oscilloscope traces showing the input and output signals of their MLP in a given configuration oscillating up and down, presumably to test the speed/throughput of the system and its power consumption. They then provide some tables of numbers summarizing measured results from other configurations. From the discussion in the results section, it is very difficult to understand precisely what they are doing and why the observed oscillatory signals are what should be expected. This reviewer might have expected to see some kind of measured transfer characteristics (or maybe even some simulated ones) of their basic building blocks (e.g. synapse, sigmoid, WTA), rather than just the overall system behavior from input to output.
The English is readable, but could definitely be improved in places.
Round 2
Reviewer 4 Report
-
Author Response
Thank you for your extensive comments in the last round. I am glad that I was able to provide all the relevant information that you requested.
Reviewer 5 Report
The authors have improved the manuscript over from the first version. Unfortunately, the draft manuscript provided for review does not have any figures included, so it is really impossible to properly review the current version of the manuscript, especially the simulated transfer characteristics that they newly included in this revision. With regard to the sub-threshold translinear multiplier, the authors did add the modified translinear principle to account for the \kappa parameter. Unfortunately, in the sentence immediately following equation (8), they make a blunder. They write "Fortunately, the \kappa term cancels out from both sides and Equation 4 still holds for the multiplier equation using weak inversion MOSFETs." That is incorrect. The \kappa here is not a multiplicative factor that can be canceled from both sides of the equation. Rather, it is a power to which one of the two currents on each side of the equation is raised. You can raise both sides of the equation to the 1 / \kappa power, which transforms the equation into I_1^(1 / \kappa) I_2 = I_3 I_4^(1 / \kappa). The \kappa does *not* go away and it is *not* sufficiently close to 1 that you can ignore it. Probably the best you can say is that its presence doesn't really matter for this application. For example, if the current raised to the \kappa power is the weight, the actual weight is given effectively by the input current raised to the \kappa power. If you want to recover the classic Gilbert translinear principle using sub-threshold MOSFETs, either you need to put each device in its own well and tie the source and body together or you need to restrict your choice of translinear loops to those that strictly alternate between clockwise and anticlockwise facing devices. This reviewer still fails to see why the differential pair analysis is included and would prefer to see the sigmoid transfer characteristic given as a function of the two input currents (i.e. the ones that flow into the diode-connected transistors that convert the currents into V_1 and V_2). That is not a hard thing to do and it makes a whole lot more sense given the way the authors are using the differential pair in their system to implement the sigmoid. This reviewer would need to see the figures and the remaining problem concerning the translinear multiplier need to be addressed before the manuscript would be suitable for publication.
additional comments:
The figures displaying the simulation results were not quite what this reviewer would have expected or hoped for. This reviewer would have expected something more like DC transfer characteristics rather than the transient simulation results provided. Why did the authors choose to provide these? They are not particularly interesting or clear. If they were to swap the synapse and sigmoid plots keeping the captions as they are, who would be the wiser? This reviewer does not believe they add much to the manuscript. The error regarding \kappa canceling must be fixed in order for the paper to be publishable.
Round 3
Reviewer 5 Report
This reviewer is fine with the current revision of the manuscript.